# An Improved OSIRIS $NO_2$ Profile Retrieval in the UTLS and Intercomparison with ACE-FTS and SAGE III/ISS

Kimberlee Dubé[1], Daniel Zawada[1], Adam Bourassa[1], Doug Degenstein[1], William Randel[2], David Flittner[3], Patrick Sheese[4], and Kaley Walker[4]

[1]Institute of Space and Atmospheric Studies, University of Saskatchewan, Saskatchewan, Canada
[2]National Center for Atmospheric Research, Boulder, CO, USA
[3]NASA Langley Research Center, Hampton, VA, USA
[4]University of Toronto, Department of Physics, Toronto, Canada

**Correspondence:** Kimberlee Dubé (kimberlee.dube@usask.ca)

**Abstract.** The v7.2 $NO_2$ retrieval for the Optical Spectrograph and InfraRed Imager System (OSIRIS) was designed to improve sensitivity in the Upper Troposphere-Lower Stratosphere (UTLS) and to reduce an observed low bias in the previous version, v6.0. The details of this retrieval are described, and then the data are compared to coincident $NO_2$ profiles from the Atmospheric Chemistry Experiment – Fourier Transform Spectrometer (ACE-FTS) and the Stratospheric Aerosol and Gas Experiment III on the International Space Station (SAGE III/ISS). The the PRATMO photochemical box model was used to account for differences in the measurement times of the instruments: all datasets were scaled to the same local solar time of 12:00 pm. Coincident ACE-FTS and OSIRIS $NO_2$ measurements agree within 20% throughout much of the stratosphere. Coincident SAGE III/ISS and OSIRIS $NO_2$ measurements also agree within 20%, with OSIRIS biased low at all altitudes and latitudes. The ACE-FTS, OSIRIS, and SAGE III-ISS $NO_2$ monthly zonal mean data show very similar variability in time at most altitude and latitudes.

## 1 Introduction

Satellite observations are crucial for monitoring changes in atmospheric composition. Measurements of stratospheric $NO_2$ in particular are important as $NO_2$ is a key factor for ozone photochemistry. It is often necessary to use data from multiple instruments in order to fully explain the distribution of $NO_2$ throughout the stratosphere, but such studies require detailed understanding of the biases between the different datasets. This can be challenging in the case of $NO_2$, where a complex daily photochemical cycle prevents the direct comparison of measurements taken at different local solar times (LSTs). Here we focus on $NO_2$ retrieved from limb scatter and solar occultation instruments. These measurements have excellent vertical resolution, making it possible to study variations in $NO_2$ from the upper troposphere to the mid-stratosphere.

Updated $NO_2$ retrievals were recently developed for several instruments: the Optical Spectrograph and InfraRed Imager System (OSIRIS, Llewellyn et al., 2004), the Atmospheric Chemistry Experiment - Fourier Transform Spectrometer (ACE-FTS, Bernath et al., 2005) and the Stratospheric Aerosol and Gas Experiment on the International Space Station (SAGE III/ISS, Cisewski et al., 2014). OSIRIS takes limb scatter measurements near 6:30 am local time (descending node), while ACE-FTS

and SAGE III/ISS use the solar occultation technique, with measurements at sunrise and sunset. The latest OSIRIS $NO_2$ retrieval, version 7.2, was designed to fix a low bias and to improve performance in the UTLS (Upper Troposphere–Lower Stratosphere) through better cloud and aerosol filtering. This retrieval is discussed in detail in Section 2. The v7.2 data is then compared to coincident $NO_2$ profiles from ACE-FTS and SAGE III/ISS. The PRATMO photochemical box model, developed by Prather and Jaffe (1990) and updated by McLinden et al. (2000) and later Adams et al. (2017), is used to account for the different instrument measurement times. The ACE-FTS and SAGE III/ISS data are described in Section 3, and the results of the comparison are given in Section 4.

## 2  The OSIRIS v7.2 $NO_2$ retrieval

### 2.1  The OSIRIS Instrument

OSIRIS has been operating from a sun-synchronous orbit on the Odin satellite since October 2001 (Murtagh et al., 2002; Llewellyn et al., 2004). The optical spectrograph measures 100 to 400 vertical profiles of limb-scattered solar irradiance each day, at wavelengths from 275 to 810 nm. Only the descending phase measurements are used here because the ascending phase measurements have inconsistent sampling due to drifts in the orbit. The equatorial crossing of the descending phase occurs near a local solar time of 6:30 am, although the exact timing varies by approximately one hour due to the spacecraft orbit.

### 2.2  Prior Data Versions

The initial OSIRIS $NO_2$ retrieval was described by Sioris et al. (2003). Subsequent versions of the retrieval were developed by Haley et al. (2004) (version 2.4), Haley and Brohede (2007) (version 3.0), Bourassa et al. (2011) ("fast" version), and Sioris et al. (2017) (version 6.0). Validation studies for previous versions of the OSIRIS retrieval accounted for the $NO_2$ daily photochemical cycle in different ways, and found mixed results. A summary of the validation results for the earlier OSIRIS $NO_2$ retrieval versions is provided in Table 1. In general the retrieved OSIRIS v3.0 and v6.0 $NO_2$ values were biased low compared to other instruments, although the bias varies with altitude.

### 2.3  v7.2 Algorithm Description

The core of the algorithm is a spectral fit to high-altitude normalized radiances in the 434.8–476.7 nm spectral region,

$$\log \frac{I_j(\lambda)}{I_{norm}(\lambda)} \sim A_j + B_j \cdot \lambda + C_j \cdot \lambda^2 + D_j \cdot \lambda^3 + E_j \cdot \sigma_{O_3}(\lambda) + y_j \cdot \sigma_{NO_2}(\lambda) \tag{1}$$

where $I_j(\lambda)$ is the OSIRIS measured radiance at tangent altitude index $j$ and wavelength $\lambda$, $\sigma_{O_3}$ and $\sigma_{NO_2}$ are the ozone and nitrogen-dioxide cross sections at the measurement tangent altitude, and $A_j$, $B_j$, $C_j$, $D_j$, $E_j$, $y_j$ are coefficients determined through a linear regression. $E_j$ and $y_j$ are related to the slant path optical depths. The ozone cross sections of Daumont et al. (1992); Brion et al. (1993); Malicet et al. (1995) are used, with the $NO_2$ cross sections taken from Vandaele et al. (1998). The forward model radiative transfer calculation includes the full temperature dependence at all altitudes and the regression for each

| NO$_2$ Version | Instrument | SZA/LST Scaling | Altitude Range | Bias | Reference |
|---|---|---|---|---|---|
| 2.4 | SAGE II v6.2 | OSIRIS to 90° | >35 km | > -25% | Brohede et al. (2007) |
|  |  |  | 25-35 km | ±20% |  |
|  |  |  | 18-25 km | > -56% |  |
|  | SAGE III/Meteor-3M v3 | OSIRIS to 90° | >35 km | ±11% | Brohede et al. (2007) |
|  |  |  | 25-35 km | ±12% |  |
|  |  |  | 18-25 km | > -21% |  |
| 3.0 | ACE-FTS v3.5 | ACE to OSIRIS LST | 25-45 km | ±10% | Sheese et al. (2016) |
|  |  |  | <25 km | <-10% |  |
| 6.0 | Mult. Balloons | OSIRIS to balloon LST | 14-37 km | ±10% | Sioris et al. (2017) |
|  | Mult. Ground-Based | All values to 12:00 pm | 12–40 km partial column | -20% | Bognar et al. (2019) |

**Table 1.** Summary of validation results for prior versions of the OSIRIS NO$_2$ retrieval. The SZA/LST scaling column describes how the different measurement times of the instruments were accounted for. A negative bias means that OSIRIS is biased low.

line of sight uses the temperature at the tangent point to compute the cross section. Both set of cross sections are sampled at the native resolution of the spectroscopic measurements (typically ∼0.02 nm) and then convolved to the OSIRIS measurement spectral resolution (1 nm). A discussion on the stability of the spectral resolution is given in Bognar et al. (2022). The cross

section temperature is that at the tangent altitude, where most of the absorption occurs, so the cross section is slightly different for each line of sight. The output of the spectral fitting is the vector $\mathbf{y} = (y_1, y_2, ..., y_m)$, which is then used to represent the observed values in the iterative equation,

$$\mathbf{x}_{i+1} = \mathbf{x}_i + \left[ \mathbf{K}^T \mathbf{S}_y^{-1} \mathbf{K} + \mathbf{R} + \gamma \, diag(\mathbf{K}^T \mathbf{S}_y^{-1} \mathbf{K}) \right]^{-1} \left[ \mathbf{K}^T \mathbf{S}_y^{-1} (\mathbf{y} - F(\mathbf{x}_i)) - \mathbf{S}_a^{-1} (\mathbf{x}_i - \mathbf{x}_a) \right], \tag{2}$$

where $\mathbf{x}$ is a vector of NO$_2$ number density on a 1 km vertical grid with length $n$, $\mathbf{K}$ is the Jacobian matrix $\partial \mathbf{y}/\partial \mathbf{x}$, $\mathbf{S}_y$

is the covariance matrix of $\mathbf{y}$, $\mathbf{x}_a$ is the apriori state, $\mathbf{R}$ is a regularization matrix, $\gamma$ is the Levenberg-Marquardt damping parameter, and $F$ is the forward model. The lowest retrieved altitude is determined from the cloud detection performed in the OSIRIS version 7 aerosol retrieval (Rieger et al., 2019), and the highest altitude extends to 40 km. The forward model is a combination of the SASKTRAN radiative transfer model (Bourassa et al., 2008; Zawada et al., 2015) and the application of Eq. 1. Included in the forward model calculation are the results from the OSIRIS v7.2 ozone, stratospheric aerosol, and surface

albedo retrievals. The measurement covariance is assumed to be diagonal and determined through the residuals of the linear regression procedure. A second-derivative Tikhonov style regularization matrix is used which is scaled by the prior state,

$$\mathbf{R} = \alpha \cdot (\mathbf{x}_a^{-1})^T \mathbf{\Gamma}^T \mathbf{\Gamma} (\mathbf{x}_a^{-1}), \tag{3}$$

where $\alpha$ is a scale factor, $\Gamma$ is a numerical second-derivative operator of size $(n-2) \times n$, and $\mathbf{x}_a^{-1}$ is the element-wise inverse of $\mathbf{x}_a$. The prior state is calculated from a latitude and month dependent climatology computed through the box model of Prather

and Jaffe (1990).

Convergence is detected through analysis of the quantity being minimized,

$$\chi_i^2 = [\mathbf{y} - F(\mathbf{x}_i)]^T S_y [\mathbf{y} - F(\mathbf{x}_i)] + [\mathbf{x}_i - \mathbf{x}_a]^T S_a [\mathbf{x}_i - \mathbf{x}_a]. \tag{4}$$

The predicted value of $\chi^2$, assuming the problem is linear, can be evaluated as,

$$\chi_{l,i}^2 = [\mathbf{y} - F(\mathbf{x}_i) - \mathbf{K}\Delta\mathbf{x}_{l,i}]^T S_y [\mathbf{y} - F(\mathbf{x}_i) - \mathbf{K}\Delta\mathbf{x}_{l,i}] + [\mathbf{x}_i - \mathbf{x}_a + \Delta\mathbf{x}_{l,i}]^T S_a [\mathbf{x}_i - \mathbf{x}_a + \Delta\mathbf{x}_{l,i}], \tag{5}$$

where $\Delta\mathbf{x}_{l,i}$ is found by setting $\gamma = 0$ in Eq. 2 and evaluating $\mathbf{x}_{i_1} - \mathbf{x}$. The iteration is then stopped when,

$$\frac{\chi_i^2}{\chi_{l,i}^2} < 1.01, \tag{6}$$

which results in profiles that have converged to a level orders of magnitude less than the estimated precision. Scans where this criteria is not achieved are flagged and discarded.

For each scan, various error characterization metrics are also calculated. The covariance of the retrieved state is calculated

as,

$$\mathbf{S}_x = \mathbf{G}\mathbf{S}_y\mathbf{G}^T, \tag{7}$$

with the gain matrix $\mathbf{G}$ given by,

$$\mathbf{G} = (\mathbf{K}^T \mathbf{S}_y^{-1} \mathbf{K} + \mathbf{R})^{-1} \mathbf{K}^T \mathbf{S}_y^{-1}. \tag{8}$$

The averaging kernel is computed through,

$$\mathbf{A} = \mathbf{G}\mathbf{K}. \tag{9}$$

To determine the optimal regularization scale factor, $\alpha$ in Eq. 3, an analysis was performed on representative OSIRIS scans. Regularization scale factor values that are too high result in averaging kernels that are not sharply peaked in the UTLS with degraded vertical resolution, while values that are too low can lead to over-fitting, oscillations, and poor convergence. Figure 1 shows an example of these tests for one OSIRIS scan and three different values of the regularization scale parameter (1, 5,

20). For $\alpha = 1$, we see large oscillations in the UTLS leading to highly negative values. At $\alpha = 20$ the oscillations are damped, however the response in the UTLS is greatly damped, with poorer vertical resolution in the stratosphere and worse agreement between the OSIRIS measurements and the forward model, particularly above 30 km. $\alpha = 5$ offers a balance between the oscillations and response, and maintains a vertical resolution of 2–3 km in most of the stratosphere which matches the vertical OSIRIS sampling resolution. For these reasons the version 7.2 processing uses a regularization value of $\alpha = 5$.

## 2.4 Comparison to Version 6.0

The version 6.0 algorithm uses a similar procedure where the measurement vector is determined from the regression fit in Eq. 1, but version 7.2 makes key improvements aimed to reduce the observed low biases and improving the knowledge of the response in the UTLS. A full description of the version 6.0 algorithm can be found in Sioris et al. (2017). The key differences between version 6.0 and version 7.2:

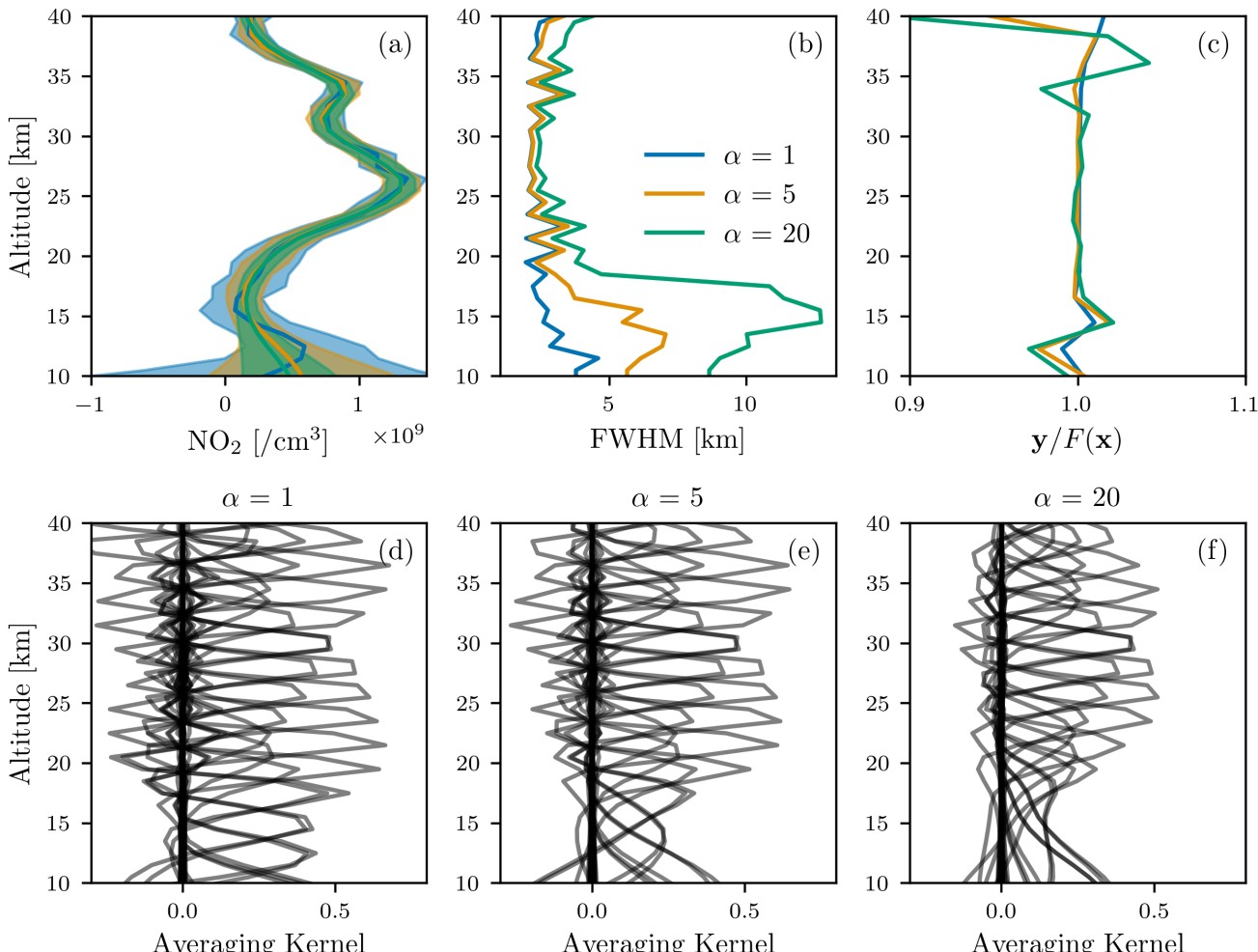

**Figure 1.** Retrieved $NO_2$ and diagnostic information for three different regularization scale values for OSIRIS scan 32921024 (March 8, 2007, 23.9°N). (a) The retrieved $NO_2$ profile (solid lines) and $1\sigma$ error estimates (shaded areas) for three regularization values. (b) The estimated vertical resolution of the retrieval for three regularization values. (c) The ratio of the OSIRIS measurement vector to the simulated measurement vector for three regularization values. (d-f) The averaging kernels for the three regularization values.

– Version 6.0 assumed pre-flight calibration values for the OSIRIS spectral resolution in the $NO_2$ absorption band. Version 7.2 fits the spectral resolution on a scan-by-scan basis by fitting to solar Fraunhofer lines. This reduces the low bias because the Full-Width Half-Maximum (FWHM) from the fitting in v7.2 is larger than the assumed FWHM in v6.0 due to the temperature of the optics decreasing over time. A wider spectral resolution results in weaker absorption features, and therefore in an increased retrieved number density to compensate.

– Version 6.0 used a fixed number of iterations of a multiplicative algebraic reconstruction technique to minimize the differences of the measurement vector. The technique forced the retrieved $NO_2$ number density to be positive, did not rigorously verify convergence, and made it computationally prohibitive to calculate averaging kernels for each scan. Version 7.2 uses Levenberg-Marquardt iteration, allows negative number densities to be retrieved, performs extensive convergence checks, and provides an averaging kernel for each scan. Negative values should be used when computing

means or the results will be biased high.

     – Version 6.0 normalized radiances from the range 50–70 km. Version 7.2 lowers the normalization range to 45–50 km in order to reduce the effect of residual straylight.

     – Both versions 6.0 and 7.2 determine the lowest retrieved altitude from cloud-detection. Version 6.0 uses the vertical gradient of radiance in the 810 nm OSIRIS measurement to detect cloud, while version 7.2 uses an improved method of

Rieger et al. (2019) that combines stratospheric aerosol information with a radiance color ratio.

Table 2 summarizes the difference between the version 6.0 and version 7.2 retrievals settings.

| Retrieval Setting | v6.0 | v7.2 |
|---|---|---|
| Iterative Method | Multiplicative Algebraic Reconstruction Technique (MART) | Levenberg-Marquardt |
| Negative values allowed? | No | Yes |
| Measurement Vector | $NO_2$ Coefficient of the spectral fit of 424.8-476.7 nm radiances | $NO_2$ Coefficient of the spectral fit of 424.8-476.7 radiances |
| Vertical Constraint | Coarse grid matching | Second order Tikhonov regularization |
| Radiative Transfer Model | SASKTRAN | SASKTRAN |
| Normalization Altitude | 50-70 km | 40-45 km |
| Cloud Filtering | 810 nm gradient | Method of Rieger et al. (2019) |
| OSIRIS Spectral Resolution Determination | Fixed from pre-flight values | Fit from solar features for each scan |
| Ancillary Data | v5 OSIRIS ozone/aerosol/albedo, ECMWF forecast pressure and temperature | v7 OSIRIS ozone/aerosol/albedo, MERRA 2 pressure and temperature |

**Table 2.** Settings for the OSIRIS v6.0 and v7.2 $NO_2$ retrievals.

     The zonal mean differences between v6.0 and v7.2 is shown in Figure 2. The v7.2 $NO_2$ concentrations are greater than the v6.0 $NO_2$, with the largest differences below 20 km. The observed differences are encouraging, suggesting that the observed low biases in v6.0 validation efforts will be reduced in v7.2. Later sections will explore this further, comparing both v6.0 and

v7.2 to co-located satellite measurements.

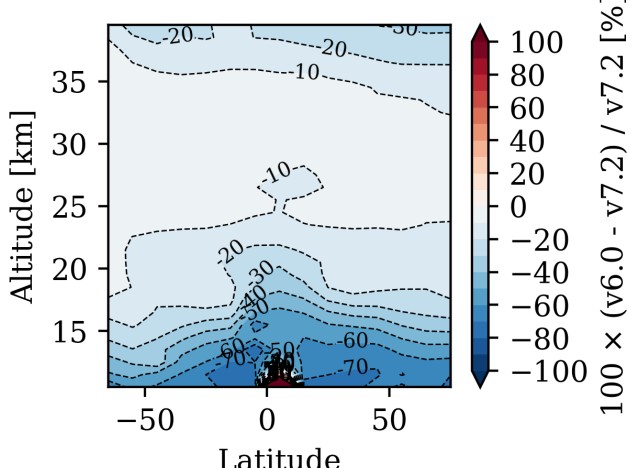

**Figure 2.** Mean percent difference between version 6.0 and version 7.2 OSIRIS $NO_2$ for all observations from November 2001 to May 2020. Bin spacing is 1 km and 10 degrees of latitude.

The second goal of v7.2 was to improve the response in the UTLS. Figure 3 shows the distributions of the v6.0 and v7.2 $NO_2$ in 20 degree latitude bins for every third altitude level below 24.5 km. The v7.2 $NO_2$ is normally distributed, but the v6.0 $NO_2$ has a log-normal distribution shape at the lower altitudes. We expect the retrieved $NO_2$ to be normally distributed because the distributions are dominated by the precision of the measurements rather than geophysical $NO_2$ variation at the lowest altitudes. Thus log-normal distributions are less physically realistic, and are likely a result of the low bias in the v6.0 retrieval, combined with the inability of the v6.0 retrieval to retrieve negative number density values.

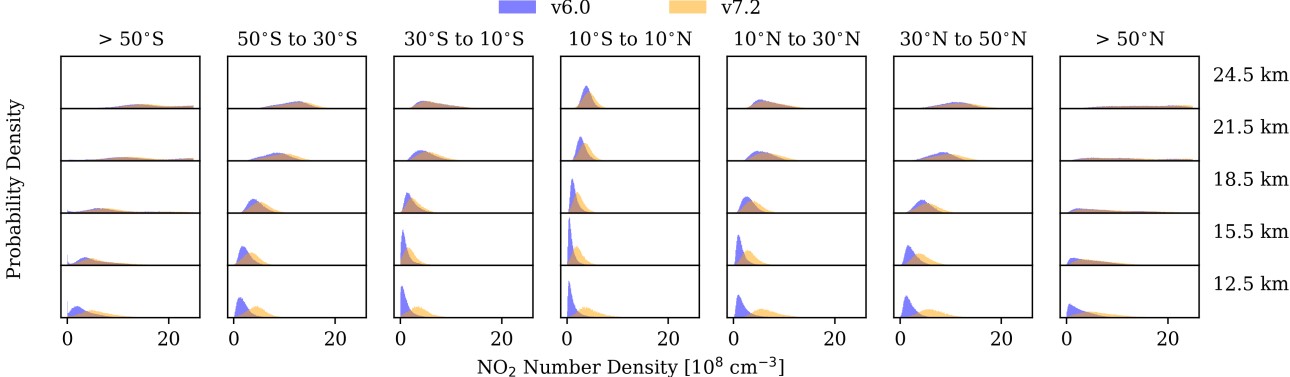

**Figure 3.** Probability densities for version 6.0 and version 7.2 OSIRIS $NO_2$ at several altitudes.

## 2.5 Averaging kernel based lower bound

The limb scatter technique rapidly loses sensitivity to $NO_2$ in the UTLS due to an increased optical path length and relatively low values of $NO_2$. The combination of the averaging kernel and retrieval error covariance matrix characterizes this loss of sensitivity, however for many scientific applications it is not possible to consider the averaging kernel directly in the analysis. For this reason a filter based on the averaging kernel for each profile was developed as a way to put a lower altitude limit on the retrieved $NO_2$. The averaging kernel $\mathbf{A}$ relates the change in the retrieved atmospheric state, $\hat{\mathbf{x}}$, to the change in the true state, $\mathbf{x}$,

$$\mathbf{A} = \frac{\partial \hat{\mathbf{x}}}{\partial \mathbf{x}}, \tag{10}$$

which characterizes the information content of the retrieval.

Ideally the averaging kernel is a sharply peaked Gaussian at the altitude for which we are retrieving information: the width of the averaging kernel defines the spatial resolution of the retrieval. This allows us to use the width of the averaging kernel and the altitude at which it peaks to characterize the performance of the retrieval.

Panel (a) of Figure 4 shows the 15.5 km and 30.5 km averaging kernels for a sample OSIRIS scan, with the Gaussian fits overlaid as dashed lines. The reported retrieval altitudes are marked with solid black lines and the peak altitudes of the Gaussian are marked with dashed black lines. The difference between these altitudes is calculated for each averaging kernel. By inspection of these differences it was determined that the filter should remove all measurements below the highest altitude at which the altitude difference is greater than or equal to 1.5 km. Several values were tested and a threshold of 1.5 km provides a compromise between including information that is far from the tangent point, and removing what are likely real geophysical signals. Panels (b), (c), and (d) of Figure 4 show the $NO_2$ profile, FWHM, and altitude difference for a sample OSIRIS profile, respectively. While there is nothing obviously unusual about the $NO_2$ itself, the FWHM increases below 15.5 km and the difference between the peak averaging kernel altitude and the reported retrieval altitude becomes greater than 1.5 km at 15.5 km. Therefore in this case the kernel filter says that the retrieval is adding minimal information below 15.5 km, and so the $NO_2$ values at lower altitudes should not be used.

The first panel of Figure 5 shows the percentage of $NO_2$ data in 2010 at each altitude and latitude that is successfully retrieved, and that is above the cloud top (OSIRIS measures scattered sunlight so it is incapable of measuring anything below the cloud top). Up to 25% of the data is retrieved down to 10 km, which is well into the troposphere in the tropics. The dashed orange line in the Figure is the average tropopause height, based on the temperature lapse rate (the value is provided with each OSIRIS profile). The solid orange line is the average 380 K potential temperature height. It was calculated using the temperature and pressure information included with the OSIRIS $NO_2$ data. This level is an alternative definition of the tropopause location. The second panel of Figure 5 shows the percentage of the data that remains after applying the averaging kernel filter to the v7.2 retrieved $NO_2$. The majority of the $NO_2$ data below the lapse rate tropopause is removed. Based on this filter, only about 20% of the $NO_2$ profiles extend down to $\sim$16 km in the tropics and $\sim$12 km at higher latitudes.

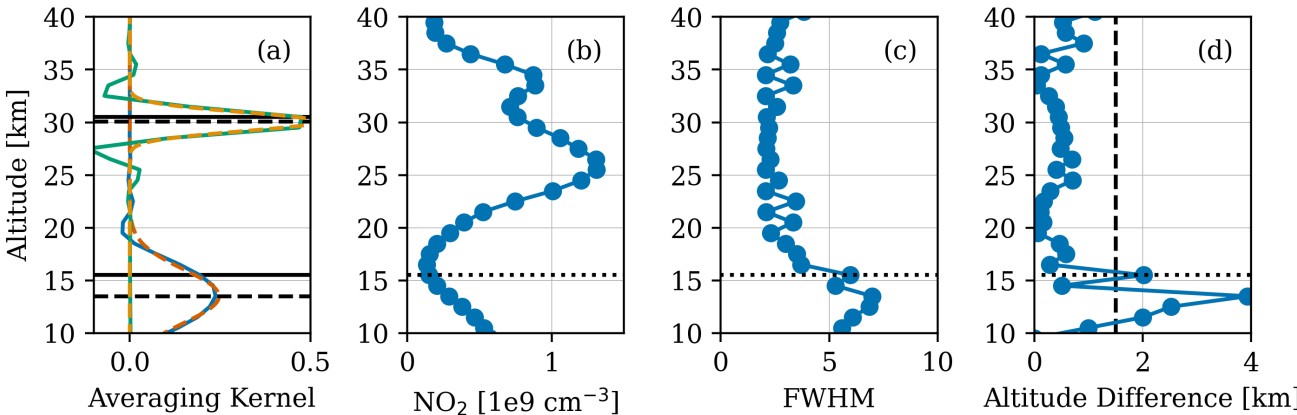

**Figure 4.** $NO_2$ retrieval diagnostics for OSIRIS scan 3292102 (March 8, 2007, 23.9°N). (a) Averaging kernels at 15.5 km (blue) and 30.5 km (green). The dashed red line is the Gaussian fit to the 15.5 km averaging kernel and the dashed line orange line is the Gaussian fit to the 30.5 km averaging kernel. The solid black lines mark the retrieval altitudes and the dashed black lines mark the peak altitude of the Gaussian fits. (b) $NO_2$ profile. (c) FWHM of the averaging kernels. (d) Difference between averaging kernel altitude and reported retrieval altitude. The dashed black line marks a difference of 1.5 km. In the last three panels the dotted black line is the altitude at which the kernel filter would cut off this profile.

## 3 Validation Datasets

### 3.1 ACE-FTS

ACE-FTS has been in orbit on SCISAT since 2003, and collecting data since February 2004. It is in a high-inclination circular orbit (74°) at 650 km. ACE-FTS is an infrared Fourier transform spectrometer, measuring from 750 cm$^{-1}$ to 4400 cm$^{-1}$, with a resolution of 0.02 cm$^{-1}$ (Boone et al., 2005, 2013). There are typically 30 occultation events each day, 15 at sunrise and 15 at sunset.

Vertical profiles for over 40 molecules and over 20 isotopologues are retrieved from the ACE-FTS measurements. The observed interferograms are first converted to atmospheric transmission spectra, and then the volume mixing ratio is retrieved from each spectrum using a non-linear least squares technique (Boone et al., 2013). The v3.5 $NO_2$ retrieval uses 40 microwindows between 1204.4 cm$^{-1}$ and 2950.9 cm$^{-1}$, and the retrieved profiles extend from a minimum altitude of 7 km to a maximum altitude of 52 km. The retrieval uses global fitting, assumes horizontal homogeneity, and does not require a priori $NO_2$ data (only a first guess). It also accounts for interfering species (e.g. $H_2O$, $CH_4$, OCS).

A detailed validation of the version 3.5 $NO_2$ retrieval is given in Sheese et al. (2016). Note that a change in the processor is the only difference between v3.5 and v3.6. The current recommended version is v4.2. The only difference between v4.1 and v4.2 is the global environment settings, which caused no significant difference between v4.1 and v4.2 $NO_2$ volume mixing ratios. v4.1 is described in Boone et al. (2020). The changes from v3.6 to v4.1/v4.2 had a minimal effect on the retrieved

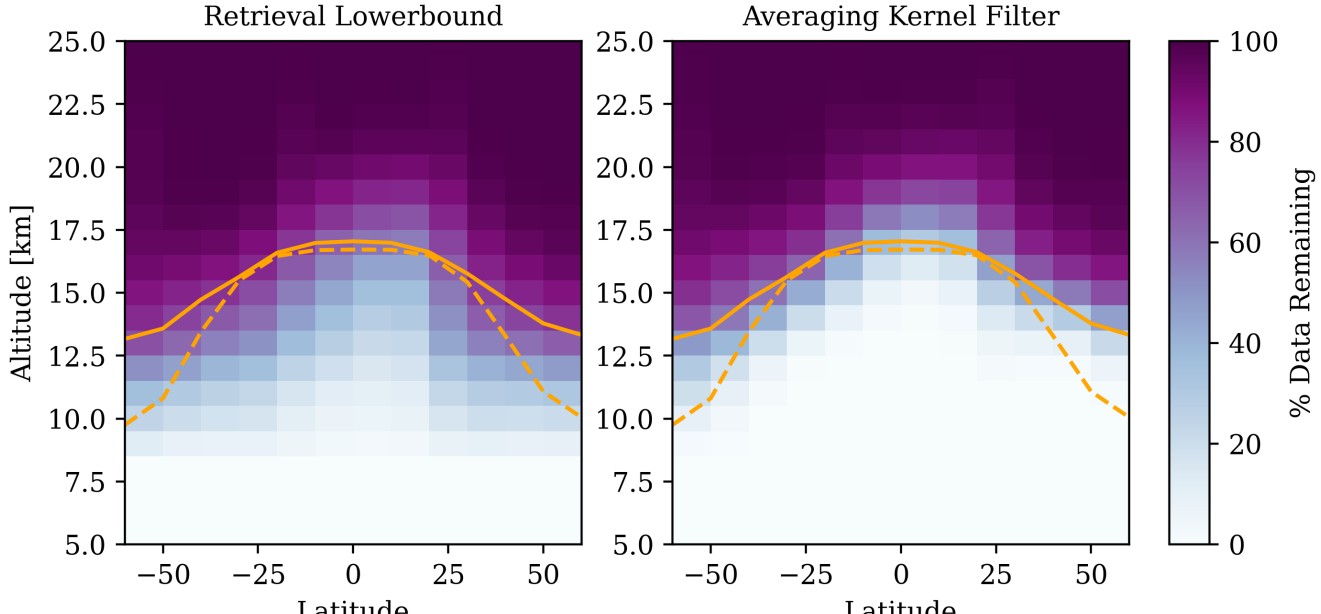

**Figure 5.** Left: The percentage of data that is retrieved and above the cloud top. Right: The percentage of data that remains after applying the averaging kernel filter. The solid orange line is the average altitude of the 380 K potential temperature. The dashed orange line is the mean tropopause height based on the temperature lapse rate. The percentages were calculated for all OSIRIS profiles from 2010.

175 $NO_2$: the difference between the two versions is within $\pm 5\%$ at most latitudes and altitudes above 15 km. Here we focus on v4.1/v4.2.

## 3.2 SAGE III/ISS

SAGE III has been collecting data from the ISS since June 2017. The inclination of the ISS is 51.6°, which allows SAGE III/ISS to view latitudes from 70° N to 70° S. It uses a configurable charge-coupled device (CCD) spectrometer, observing

180 wavelengths from 280 nm to 1035 nm, with a 1–2 nm resolution. A separate photodiode observes from 1542 nm $\pm$ 15 nm. SAGE III/ISS continuously scans back and forth across the sun during each occultation in order to measure the irradiance as a function of altitude. There are typically 16 sunrise and 16 sunset events per day.

The measured irradiances are used to determine the $O_3$, $NO_2$, and $H_2O$ number densities, along with the aerosol extinction at several wavelengths. The algorithm first uses the measured irradiance to calculate slant path transmission profiles for each

185 channel. Each slant path transmission profile is converted to a slant path optical depth profile containing contributions from Rayleigh scattering, aerosol extinction, and absorption by at least one species. Multiple linear regression is then used to solve for the $NO_2$ and $O_3$ slant path number density profiles simultaneously. $NO_2$ is retrieved from channel S3 (433-450 nm). A global fit method is used to convert the slant path number density to vertical number density profiles. Further details on the

retrieval are given in the SAGE III Algorithm Theoretical Basis Document (2002). The $NO_2$ number density is available from 10 to 45 km on a 0.5 km grid with a vertical resolution of around 1.5 km. The reported uncertainty due to measurement noise in the SAGE III/ISS $NO_2$ is approximately 5% at 30 km, increasing up to 20% at 10 and 40 km.

The recent v5.2 retrieval algorithm improves upon the v5.1 algorithm in many aspects, the most important being refined wavelength map and bandpass for the spectrograph. Additional improvements relevant to $NO_2$ include better oxygen dimer ($O_4$) corrections and the removal of all vertical smoothing of the input Level 1 transmission profiles. In addition, the number density profiles are not smoothed in v5.2, as they were in v5.1. A five-point triangular smoothing was applied to each individual profile used here in order to better compare with v5.1. This smoothing is comparable to the 2-3 km vertical resolution provided by OSIRIS.

### 3.2.1 The Diurnally Varying Retrieval

Photochemistry causes the $NO_2$ number density to vary throughout the course of a day. During an occultation measurement the solar zenith angle (SZA) is 90° at the tangent point, but it varies along the line of sight (LOS). The SAGE III/ISS and ACE-FTS $NO_2$ retrieval algorithms both neglect these deviations along the instrument's LOS by assuming there is a constant gradient in the $NO_2$ number density with respect to the vertical dimension within each layer of the atmosphere. This assumption can result in retrieved $NO_2$ that is biased high at the tangent point.

Dubé et al. (2021) describes an update to the SAGE III/ISS retrieval that accounts for variations in $NO_2$ along the LOS (referred to as the diurnally varying (DV) retrieval). They used the $NO_2$ number densities from the SAGE v5.1 retrieval. The $NO_2$ values at each point along the LOS for a given scan were scaled to the SZA at that location using factors calculated with the PRATMO photochemical box model (Prather and Jaffe, 1990; McLinden et al., 2000). The input to PRATMO is an atmospheric state, consisting of pressure, temperature, air density, and $O_3$ profiles. These values are set to be those provided with the SAGE III/ISS $NO_2$ data. The sensitivity of the PRATMO $NO_2$ to the exact values of the input parameters was estimated by perturbing them in the model. The effect on $NO_2$ is small, with $NO_2$ being most sensitive to changes in temperature: a variation on the order of -1°K results in a 1% change in $NO_2$. The PRATMO inputs are kept constant as the model iterates over a set of chemical reactions for a single day. This continues until the start and end values converge. The result is a 24-hour steady state system of each species in the model. This allows us to get the $NO_2$ number density at any specified SZA.

Dubé et al. (2021) found that accounting for diurnal variations in the SAGE III/ISS retrieval improved agreement between SAGE III/ISS and OSIRIS $NO_2$ by up to 20% below 25 km. This DV retrieval, applied to both SAGE v5.1 and v5.2 $NO_2$ products, is considered in the comparisons with OSIRIS presented here, along with the standard SAGE v5.1 and v5.2 retrievals.

### 4 Intercomparison

The coincidence criteria are 1 day, 5° latitude, and 10° longitude. Figure 6 shows the number of coincident profiles with OSIRIS in each 10° latitude bin for both ACE-FTS and SAGE III/ISS. The ACE-FTS orbit results in significantly more coincidences at the high latitudes. The relatively low number of coincidences between OSIRIS and SAGE III/ISS is due to the much shorter

overlap period between the missions, compared to OSIRIS and ACE-FTS. Most latitude bins still have at least 100 pairs. The lack of coincidences with SAGE III/ISS from 0 to 30 degrees in the Northern hemisphere is because OSIRIS took few measurements in this region during 2019 and 2020, which makes up the bulk of the overlap with the SAGE III/ISS mission.

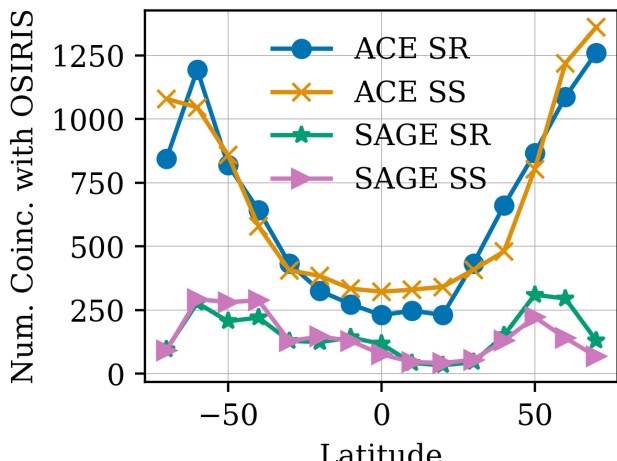

**Figure 6.** Number of coincident profiles in each $10°$ latitude bin for ACE-FTS and SAGE III/ISS with OSIRIS at 25.5 km. SS refers to sunset occultations, while SR refers to sunrise occultations.

The daily photochemical cycle results in considerably different $NO_2$ concentrations at sunrise and sunset, when ACE-FTS and SAGE III/ISS measure, and near 6:30 am, when OSIRIS measures. This must be accounted for before the different datasets can be compared. All datasets were shifted to a common local solar time of 12:00 pm using PRATMO. The ratio of the model $NO_2$ at 12:00 pm to the model $NO_2$ at the instrument measurement time is used to scale the measured $NO_2$ to 12:00 pm. This method is further described in Dubé et al. (2020).

While this scaling generally works well, it cannot always account for the differences between sunrise and sunset occultations from a single instrument. As an example, Figure 7 compares the OSIRIS and ACE-FTS $NO_2$ distributions at three altitudes in the tropics (left), at mid-latitudes (centre), and at high latitudes (right). The labels OSIRIS SR and OSIRIS SS refer to OSIRIS coincidences with ACE-FTS sunrise and sunset occultations, respectively. After scaling to 12:00 pm the sunrise and sunset distributions in the tropics and mid-latitudes have similar shapes, but a bias in the mean values. At high latitudes there is a clear double peak structure in the ACE-FTS sunset measurements, and corresponding OSIRIS coincident data. This shape is caused by the time of year at which the measurements are taken. The sunrise coincidences are mostly from the NH summer months, but the sunset coincidences also include NH spring observations. The sunrise distribution at these latitudes has a different shape than the sunset distribution. One should be cautious of this difference in shape and the bias between the mean values if intending to combine sunrise and sunset $NO_2$ data from ACE-FTS (or SAGE III/ISS). In order to avoid any such complications we consider sunrise and sunset occultations separately throughout this work.

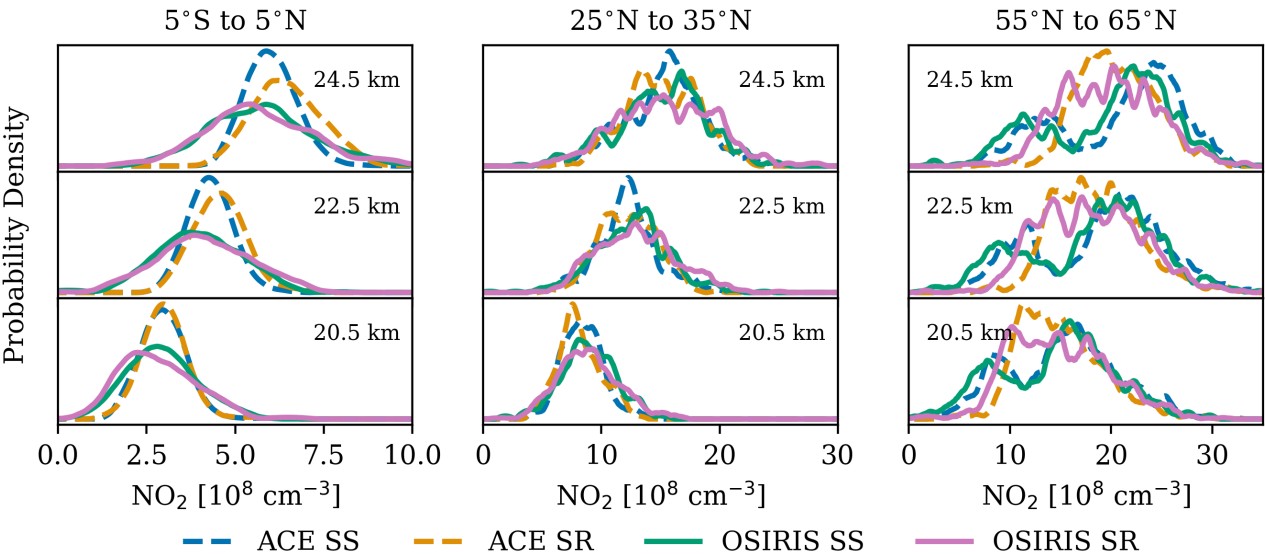

**Figure 7.** Probability densities for ACE-FTS v4.1 and OSIRIS v7.2 coincident $NO_2$, after shifting from their individual local times to 12:00 pm, for several altitudes from 5°S to 5°N (left) and from 55°N to 65°N (right).

## 4.1 Comparison with ACE-FTS

As an example, Figure 8 shows coincident ACE-FTS and OSIRIS $NO_2$ profiles from 5° to 15° latitude. This bin is representative of the difference profile structure in the Northern hemisphere. OSIRIS v7.2 shows better agreement with ACE-FTS than v6.0 above 20 km. As expected, there is minimal difference in the $NO_2$ from the two ACE-FTS retrievals. The percent difference between OSIRIS and ACE-FTS is of comparable magnitude at both sunrise and sunset, except at the highest altitudes where OSIRIS v7.2 agrees better with the ACE sunset data. The cloud and averaging kernel filters result in OSIRIS v7.2 having fewer data points than v6.0 below ~20 km, which could be why v6.0 shows better agreement with ACE-FTS at the lower altitudes.

Figure 9 shows the mean percent difference between coincident ACE-FTS and OSIRIS $NO_2$ profiles for several versions of the OSIRIS retrieval. This figure only considers the ACE-FTS v4.1 retrieval as the difference from the v3.6 retrieval is minimal. The first column compares OSIRIS v6.0 to ACE-FTS. OSIRIS is biased low everywhere, with the most negative bias occurring at lower altitudes. The middle column compares OSIRIS v7.2 to ACE-FTS. OSIRIS is lower than ACE-FTS in the Southern hemisphere below 30 km, and higher than ACE-FTS above 30 km. The lower bias in the Southern hemisphere appears in the comparisons with both OSIRIS v6.0 and v7.2 $NO_2$ so it is likely a feature of the ACE-FTS $NO_2$ product. In the Northern hemisphere the OSIRIS profiles coincident with ACE-FTS have a higher mean $NO_2$ value than the profiles from the full OSIRIS mission.

For the most part the difference between ACE-FTS and OSIRIS v7.2 is less than the difference between ACE-FTS and OSIRIS v6.0. The only regions this is not true are for the sunrise measurements (top row) above 33 km, and for both sunrise

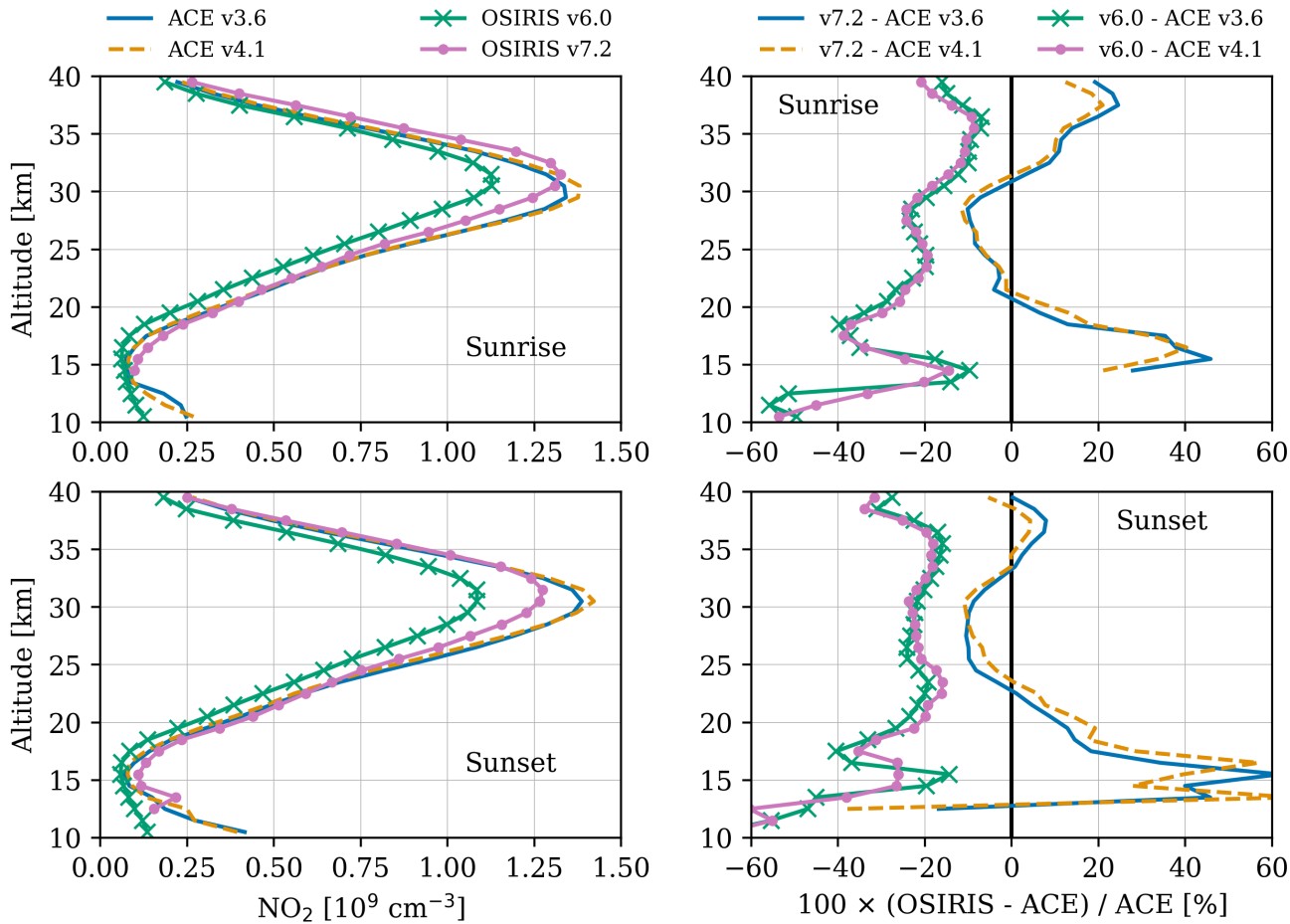

**Figure 8.** Comparison of mean coincident ACE-FTS and OSIRIS NO$_2$ profiles from 5° to 15° latitude. Top: ACE-FTS sunrise occultations. Bottom: ACE-FTS sunset occultations.

and sunset in the troposphere. Including the averaging kernel filter for the OSIRIS data reduces the bias with ACE-FTS sunset occultations in the troposphere by up to ∼50% from the v7.2 difference without the filter.

## 4.2 Comparison with SAGE III/ISS

Figure 10 compares coincident SAGE III/ISS and OSIRIS NO$_2$ profiles from -25° to 15° latitude. This sample bin generally represents the differences between SAGE III/ISS and OSIRIS in the Southern hemisphere. OSIRIS is biased lower than SAGE III/ISS at all altitudes for each retrieval version included in the figure. The best agreement occurs between OSIRIS v7.2 and SAGE III/ISS DV v5.2 (the most recent version for each instrument). Including the diurnal correction reduces the SAGE III/ISS NO$_2$ by about 10% at 25 km and more than 10% at lower altitudes (see Figure 7 of Dubé et al. (2021)). For the most part the magnitudes of the differences between OSIRIS v7.2 and SAGE III/ISS are comparable for both sunrise and sunset occultations.

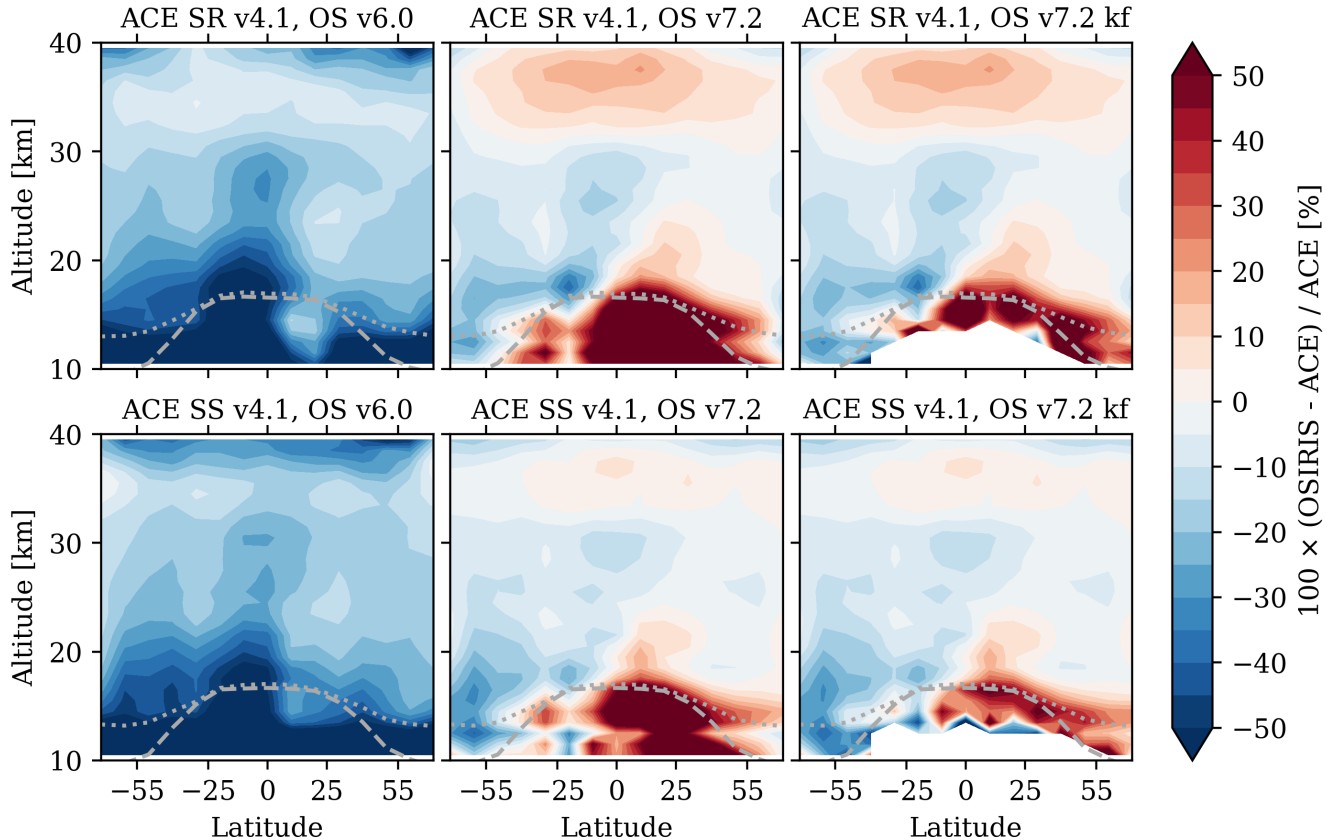

**Figure 9.** Mean percent difference between coincident profiles from ACE-FTS v4.1 and OSIRIS. Top row: Sunrise occultations. Bottom row: Sunset occultations. The last column is the same as the centre column, but with the averaging kernel filter applied to the OSIRIS $NO_2$. In all panels the dashed line is the average tropopause altitude and the dotted line is the average 380 K potential temperature altitude.

Figure 11 shows the mean percent difference between coincident SAGE III/ISS and OSIRIS $NO_2$ profiles for the v5.2 diurnally varying SAGE III/ISS retrieval. The difference between SAGE III/ISS and OSIRIS is smaller for version 7.2 of the OSIRIS retrieval than for version 6.0 at all latitudes above 20 km. The agreement is better with sunrise, rather than sunset, occultations at the higher altitudes (which is the opposite of what we see when comparing to ACE). The averaging kernel filter removes some of the values with a very high difference at the low altitudes. For the sunset $NO_2$ there is still a region remaining where OSIRIS is biased quite high. In this region the diurnally varying retrieval has a large effect, resulting in much lower SAGE III/ISS data. This increases the bias with OSIRIS, compared to using the standard SAGE III/ISS retrieval (discussed in Dubé et al. (2021)).

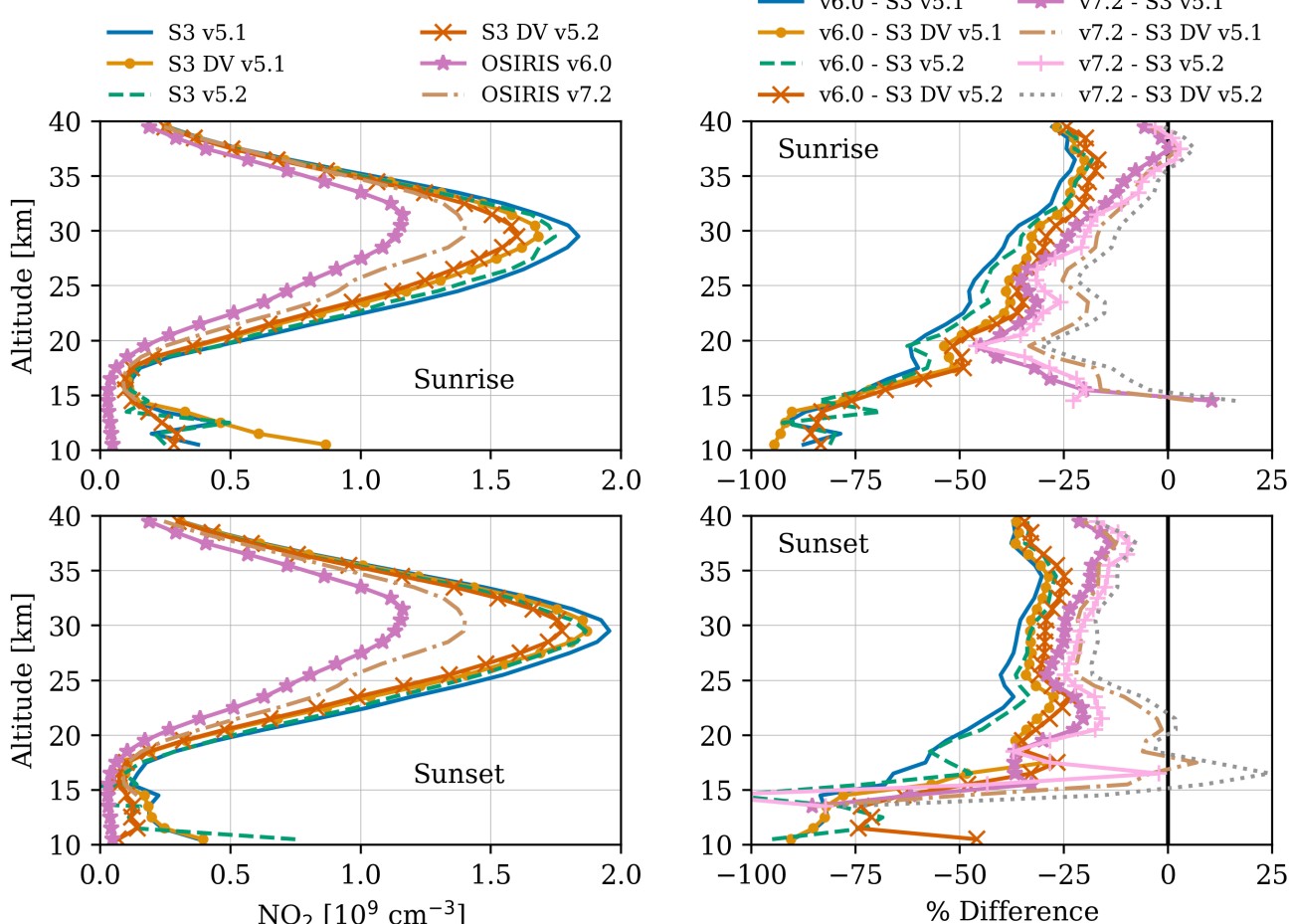

**Figure 10.** Comparison of mean coincident SAGE III/ISS and OSIRIS NO$_2$ profiles from -25° to -15° latitude. Top: SAGE sunrise occultations. Bottom: SAGE III/ISS sunset occultations.

## 4.3 The Diurnal Effect

Neither the OSIRIS retrieval nor the ACE-FTS retrieval accounts for diurnal variations in NO$_2$ along the instrument's LOS. This effect can result in a significant bias in the data below ∼25 km. Here we estimate the effect of diurnal variations on the percent difference between NO$_2$ from OSIRIS and ACE-FTS, and OSIRIS and SAGE III/ISS. Note that the magnitude of the bias caused by neglecting the diurnal effect is expected to be the same in all versions of the OSIRIS and ACE-FTS NO$_2$ data.

The magnitude of the diurnal effect depends on the direction of the LOS relative to the sun. For occultation instruments the SZA is always 90°, but for limb scatter instruments the SZA is not consistent. McLinden et al. (2006) examined the effect of diurnal variations in retrievals of NO$_2$ from OSIRIS and found that neglecting the diurnal effect can introduce a bias of 10% to 35% in the lower stratosphere for ∼16% of the NO$_2$ profiles. The bias is greater the closer the SZA is to 90°. Based on the

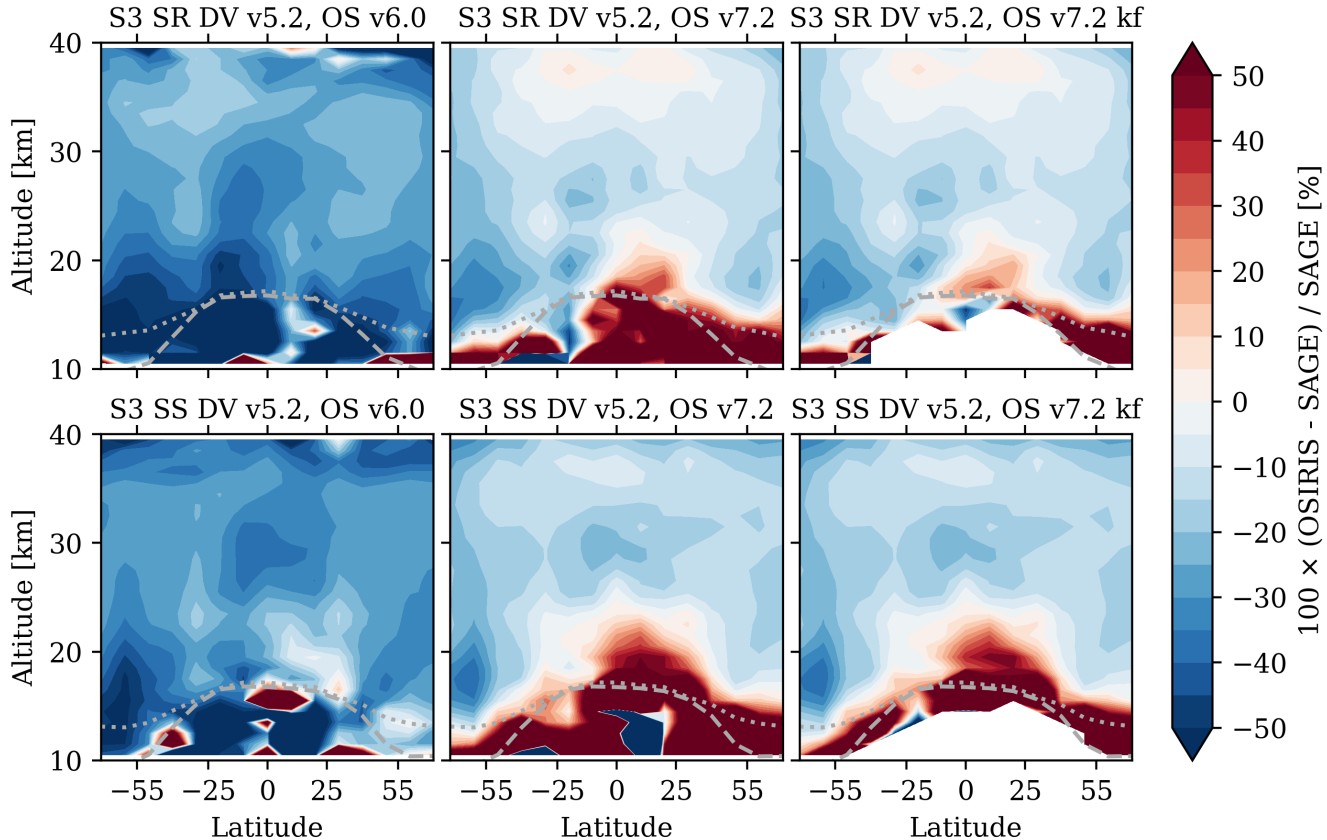

**Figure 11.** Mean percent difference between coincident profiles from SAGE III/ISS and OSIRIS. Top row: Sunrise occultations. Bottom row: Sunset occultations. The last column is the same as the centre column, but with the averaging kernel filter applied to the OSIRIS NO$_2$. In all panels the dashed line is the average tropopause altitude and the dotted line is the average 380 K potential temperature altitude.

findings of McLinden et al. (2006) we repeated our analysis using only OSIRIS scans that have a SZA $< 85°$. This reduced

285    the number of coincidences with ACE-FTS by 14% and the number of coincidences with SAGE III/ISS by 12%. The majority

of the removed coincidences were in the Southern Hemisphere. Figure 12 shows the change in the bias between OSIRIS and

ACE-FTS and OSIRIS and SAGE III/ISS after excluding these coincident profiles. In the Figure a negative number means

that removing OSIRIS scans with a significant diurnal effect improved the agreement (reduced the bias). In general the only

significant changes to the mean percent difference between ACE-FTS and OSIRIS NO$_2$ occur below the tropopause. For SAGE

290    III/ISS the difference is also small in most bins above the tropopause, although it does decrease by up to 10% in some bins in

the lower stratosphere. This bias with the SAGE III/ISS sunset occultations also decreases noticeably at all altitudes near -50°,

which is where the majority of the removed OSIRIS coincidences occur.

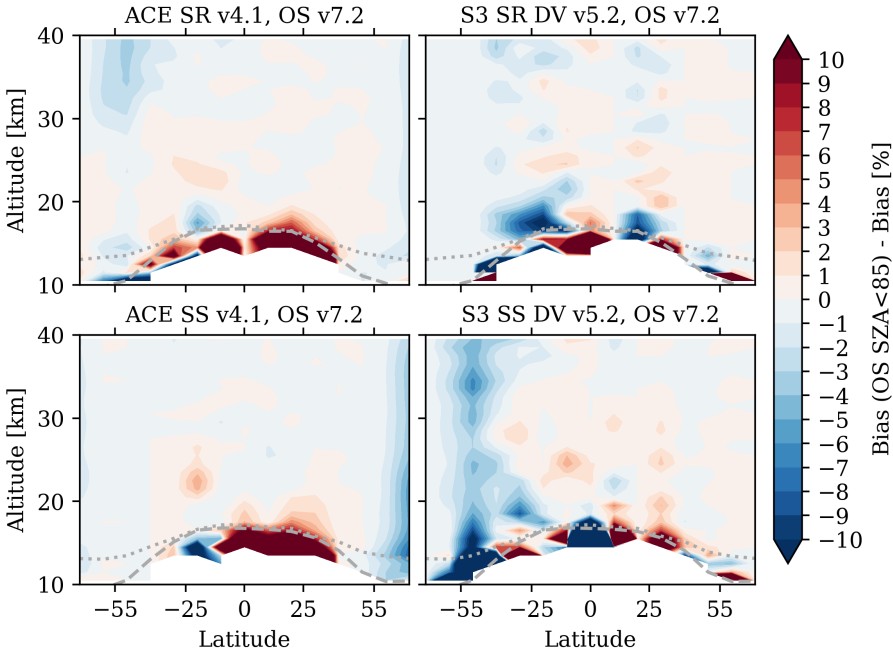

**Figure 12.** Difference between the bias with and without including OSIRIS scans with SZA $< 85°$. Top left: ACE-FTS sunrise occultations. Top right: SAGE III/ISS sunrise occultations. Bottom left: ACE-FTS sunset occultations. Bottom right: SAGE III/ISS sunset occultations. In all panels the dashed line is the average tropopause altitude and the dotted line is the average 380 K potential temperature altitude.

It is also necessary to consider the effect of diurnal variations in the ACE-FTS $NO_2$. Based on the results of Dubé et al. (2021) for SAGE III/ISS and of Brohede et al. (2007) for a simulated occultation instrument, we expect there to be a high bias of greater than 10% in the ACE $NO_2$ below $\sim$30 km. Accounting for this effect would improve the agreement with OSIRIS in the SH, but decrease agreement in the NH where ACE is already low compared to OSIRIS.

## 4.4 Time Series Comparison

Figure 13 shows the monthly mean relative anomaly time series for OSIRIS v7.2, SAGE III/ISS v5.2 DV, and ACE-FTS v4.1 $NO_2$ in several bins. The relative anomaly is calculated by subtracting the overall mean value for a given month from all values for that month to remove the seasonal cycle (e.g. the mean June $NO_2$ concentration is subtracted from each individual June $NO_2$ concentration), and then dividing by the overall mean of the data. The relative anomaly allows for sources of variability, apart from the seasonal cycle, to be more easily detected. The bins were chosen to show a range of latitudes and altitudes, with a focus on the lower altitudes as this is where OSIRIS v7.2 $NO_2$ changed most from the previous version. The anomaly time series' for several more bins are provided in Appendix A. Overall the datasets shows similar variability over a range of altitudes and latitudes. The correlation of OSIRIS with ACE-FTS sunset $NO_2$ is greater than 0.7 at most latitudes from 20 to

35 km, and the correlation with ACE-FTS sunrise $NO_2$ is greater than 0.5 at most latitudes. The correlation of OSIRIS with SAGE III/ISS is slightly lower, but still greater than 0.5 at most altitudes above 20 km from -40° to 40°. The lower correlation is likely because there are only a few years of SAGE III/ISS data available.

The ACE-FTS sunrise $NO_2$ is noisier than the sunset data in the top panel of Figure 13 (low altitude/high latitude). Sheese et al. (2016) suggested that this is because of differences in the diurnal variation along the LOS between sunrise and sunset observations. At sunrise ACE-FTS samples a region of the atmosphere that has not been illuminated long enough for the $NO_2$ to reach equilibrium, however this is not an issue at sunset. If this was indeed the problem it should also affect the SAGE III/ISS sunrise $NO_2$, but that is not the case. In addition, the SAGE III/ISS anomaly time series looks very similar, whether or not the diurnal variations along the LOS are included in the retrieval.

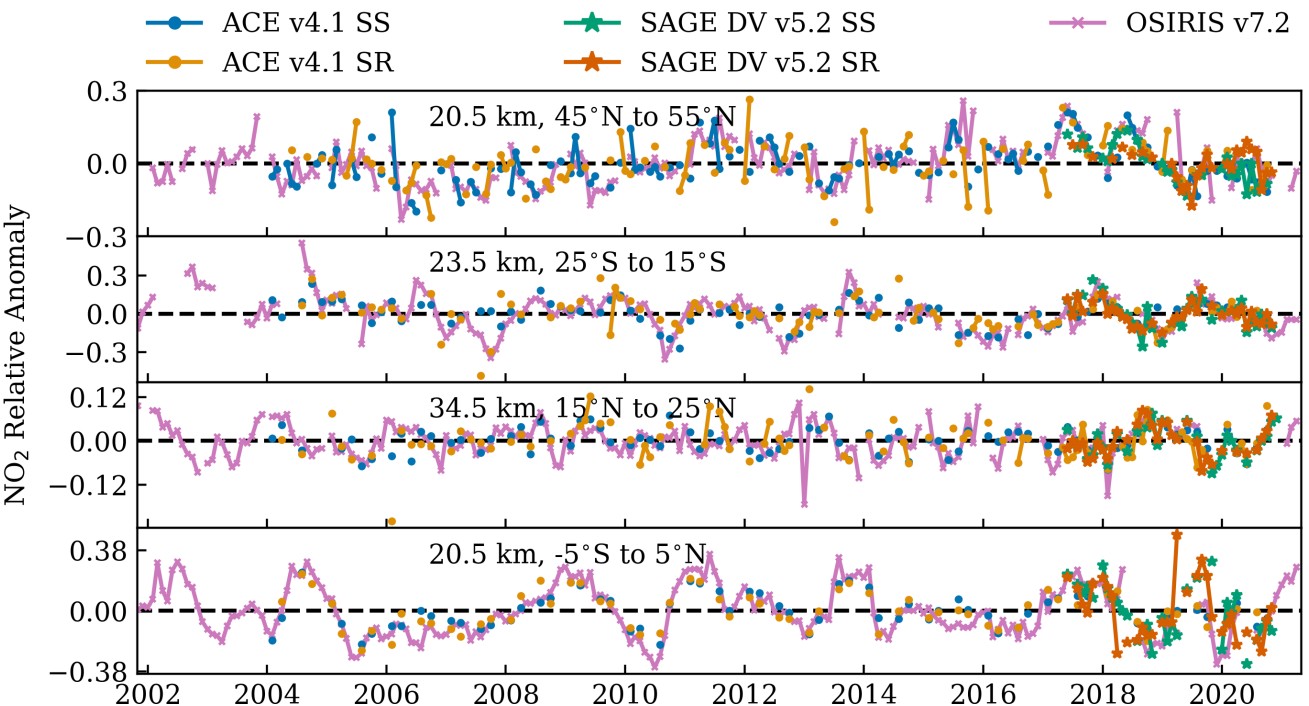

**Figure 13.** Anomaly time series comparing OSIRIS v7.2 to ACE v4.1 and SAGE III/ISS v5.2 DV for four latitude/altitude bins.

There is significant variability within each panel of Figure 13. Dubé et al. (2020) merged the previous OSIRIS v6.0 $NO_2$ with $NO_2$ from SAGE II and found that elevated aerosol levels and the quasi-biennial oscillation were the main factors influencing the $NO_2$ anomaly. Dubé et al. (2020) also showed that there is a significant increasing trend in $NO_2$ of 8-10% in the tropical lower stratosphere from 1984 to 2014.

## 5 Conclusions

A new version of the OSIRIS $NO_2$ retrieval was developed with the goal of reducing an observed low bias in the previous OSIRIS $NO_2$ version, and improving the retrieval response in the UTLS. The major improvements are: better knowledge of the OSIRIS spectral resolution, attempts to reduce the effect of residual straylight, a different iterative scheme to improve convergence, and better cloud filtering. The improved spectral fitting and the lowering of the normalization altitude to reduce stray light are the main factors that result in higher retrieved $NO_2$ number densities. A filter based on the averaging kernel was

also developed, as a way to determine that lowest altitude at which the retrieved $NO_2$ contains useful information. The values for this filter are provided in the OSIRIS v7.2 data files. This new OSIRIS v7.2 $NO_2$ retrieval was compared to coincident profiles from two occultation instruments: ACE-FTS and SAGE III/ISS. PRATMO was used to scale all datasets to 12:00 pm local time to account for the diurnal cycle in $NO_2$ before performing the comparisons. OSIRIS v7.2 agrees better with $NO_2$ from both ACE-FTS and SAGE III/ISS than the previous OSIRIS v6.0. The agreement is within 20% at most latitudes and

altitudes. In general OSIRIS agrees better with ACE-FTS than with SAGE III/ISS, although this could be due to the higher number of coincidences with ACE-FTS.

  OSIRIS agrees slightly better with ACE-FTS sunset, rather than sunrise, occultations. The bias between OSIRIS v7.2 and ACE-FTS sunset is within 10% above 20 km. This is likely because the ACE-FTS sunrise data is noisier. Conversely, OSIRIS agrees slightly better with SAGE III/ISS sunrise occultations, rather than sunset. The average sunset profile has a higher peak

$NO_2$ number density than the average sunrise profile, resulting in a greater bias with OSIRIS. As noted, the photochemical scaling to 12:00 pm is not able to account for all the differences between measurements taken at sunrise and sunset, when there are considerable changes occurring in the nitrogen chemistry.

  $NO_2$ from the SAGE III/ISS DV retrieval shows improved agreement with OSIRIS compared to $NO_2$ from standard SAGE retrieval. The diurnal effect produces a high bias in $NO_2$ retrieved from occultation instruments below 25 km. There is no

version of the ACE-FTS $NO_2$ retrieval that accounts for diurnal variations along the line of sight, which could be increasing the difference with OSIRIS below about 25 km. It is not expected that the diurnal effect would be greater in ACE-FTS than in SAGE III/ISS, so this will add at most a 5% to 40% bias, with the largest bias at the lowest altitudes (Dubé et al., 2021). If we assume an approximate high bias of 25% below 20 km in the ACE-FTS $NO_2$, the difference between ACE-FTS and OSIRIS will become greater near the tropopause, and especially in the NH. The bias in this region will be >50%, as it is for

SAGE III/ISS and OSIRIS. In the SH the bias between ACE-FTS and OSIRIS is ∼-15% below 20 km. In this region including the diurnal correction will result in a bias of ∼10%, corresponding to improved agreement between OSIRIS and ACE-FTS. This is opposite in sign and smaller in absolute value than the bias between SAGE III/ISS and OSIRIS in this region. There are significantly more coincidences between ACE-FTS and OSIRIS than there are between SAGE III/ISS and OSIRIS at SH mid-high latitudes, which could explain why we see a smaller bias between ACE-FTS and OSIRIS.

We also considered the effect of diurnal variations along the OSIRIS LOS by repeating the analysis using only OSIRIS scans with a SZA <85°, corresponding to scans shown by McLinden et al. (2006) to have a significant bias. Removing these scans had a minimal effect on the comparisons between OSIRIS and ACE-FTS in the stratosphere, changing the bias by at most 3%.

It had a greater effect on the comparison with SAGE III/ISS, where the agreement improved by up to 10% in some bins in the lower stratosphere.

The anomaly time series from each dataset shows very similar variability. Both ACE-FTS and OSIRIS are aging so it will soon be necessary to use a newer instrument like SAGE III/ISS to extend the $NO_2$ data record. The good agreement between the time series' provides confidence that SAGE III/ISS $NO_2$ can easily be combined with $NO_2$ from OSIRIS and/or ACE-FTS in the same manner that Dubé et al. (2020) combined $NO_2$ from SAGE II (the precursor to SAGE III/ISS) and OSIRIS. These long-term datasets are important for monitoring the trend in $NO_2$ as increasing anthropogenic $N_2O$ emissions are the

associated increase in stratospheric $NO_2$ can result in a decrease of $O_3$.

Overall, we conclude based on comparison with $NO_2$ from ACE-FTS and SAGE III/ISS that the OSIRIS v7.2 $NO_2$ product is an improvement over the previous v6.0 OSIRIS $NO_2$, particularly below 30 km. The bias between OSIRIS v7.2 and both ACE-FTS and SAGE III/ISS has decreased from v6.0, except for in a few bins near the tropopause where including the diurnal effect in SAGE III/ISS (or estimating the magnitude of the effect for ACE-FTS) makes the bias greater than it was in v6.0

*Data availability.*   OSIRIS data are available at https://research-groups.usask.ca/osiris/data-products.php#OSIRISLevel2DataProducts.
SAGE III/ISS data are available at https://asdc.larc.nasa.gov/project/SAGE%20III-ISS/g3bssp_52.
SAGE III/ISS $NO_2$ corrected for diurnal variations is available at https://research-groups.usask.ca/osiris/data-products.php/#SAGEIII
ACE data are available at https://databace.scisat.ca/l2signup.php.

*Code and data availability.*   The PRATMO python package can be downloaded from https://arg.usask.ca/wheels/
Documentation is available at https://arg.usask.ca/docs/Pratmo/pratmo_boxmodel.html

## Appendix A: Anomaly Time Series Figures

*Author contributions.*   K.D. performed the analysis and prepared the manuscript. D.Z. created the OSIRIS v7.2 $NO_2$ retrieval. A.B. and

D.D. assisted with the analysis and the creation of the OSIRIS data. D.F. provided guidance on using the SAGE III/ISS data. P.S. provided guidance on using the ACE-FTS data. All authors provided significant feedback on the analysis and the manuscript.

*Competing interests.*   The authors declare that they have no conflicts of interest.

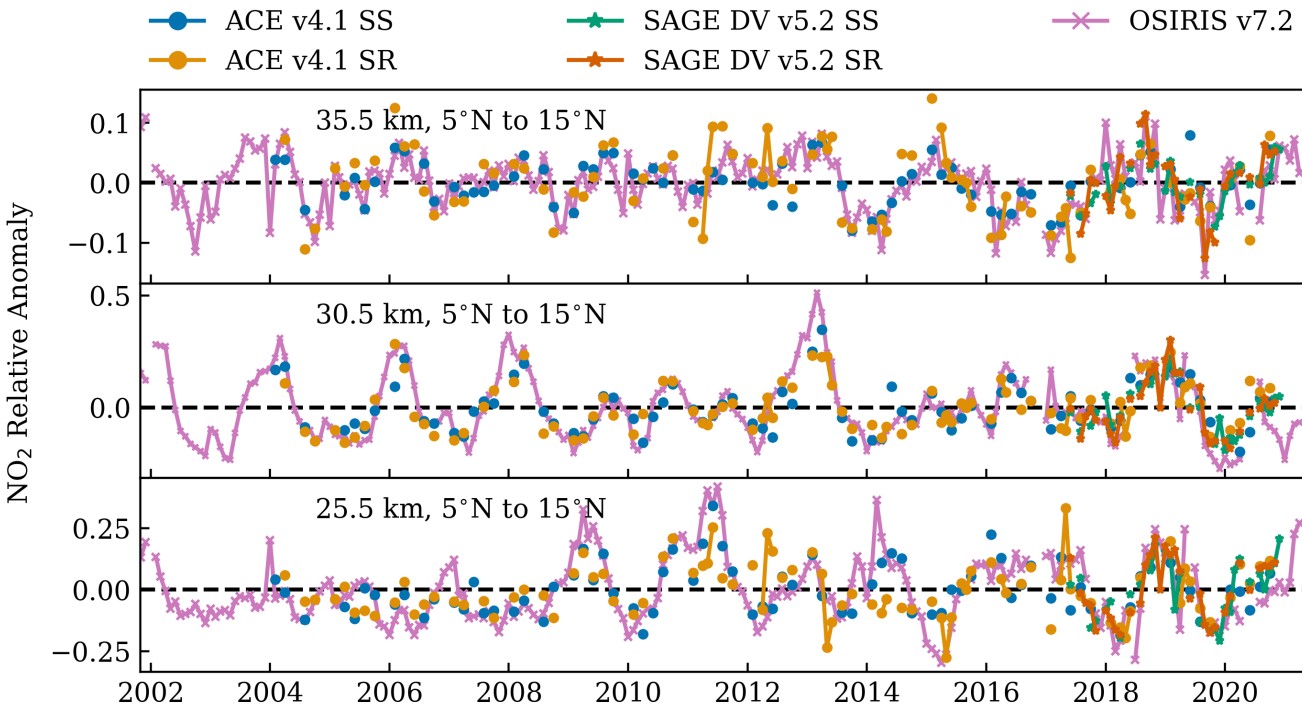

**Figure A1.** Anomaly time series comparing OSIRIS v7.2 to ACE v4.1 and SAGE III/ISS v5.2 DV for three altitudes in the tropics.

*Acknowledgements.* The authors thank the Swedish National Space Agency and the Canadian Space Agency for the continued operation and support of Odin-OSIRIS. The Atmospheric Chemistry Experiment (ACE) is a Canadian-led mission mainly supported by the CSA and the NSERC, and Peter Bernath is the principal investigator.


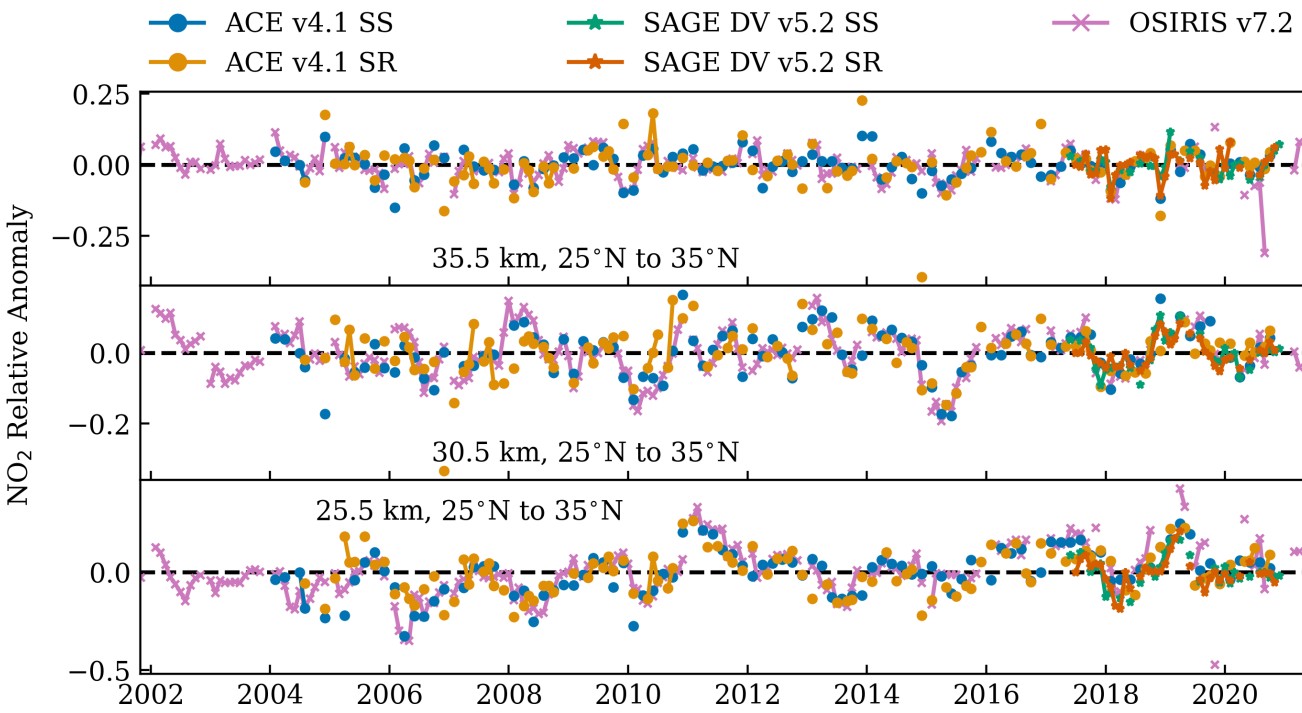

**Figure A2.** Anomaly time series comparing OSIRIS v7.2 to ACE v4.1 and SAGE III/ISS v5.2 DV for three altitudes at NH subtropics.

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

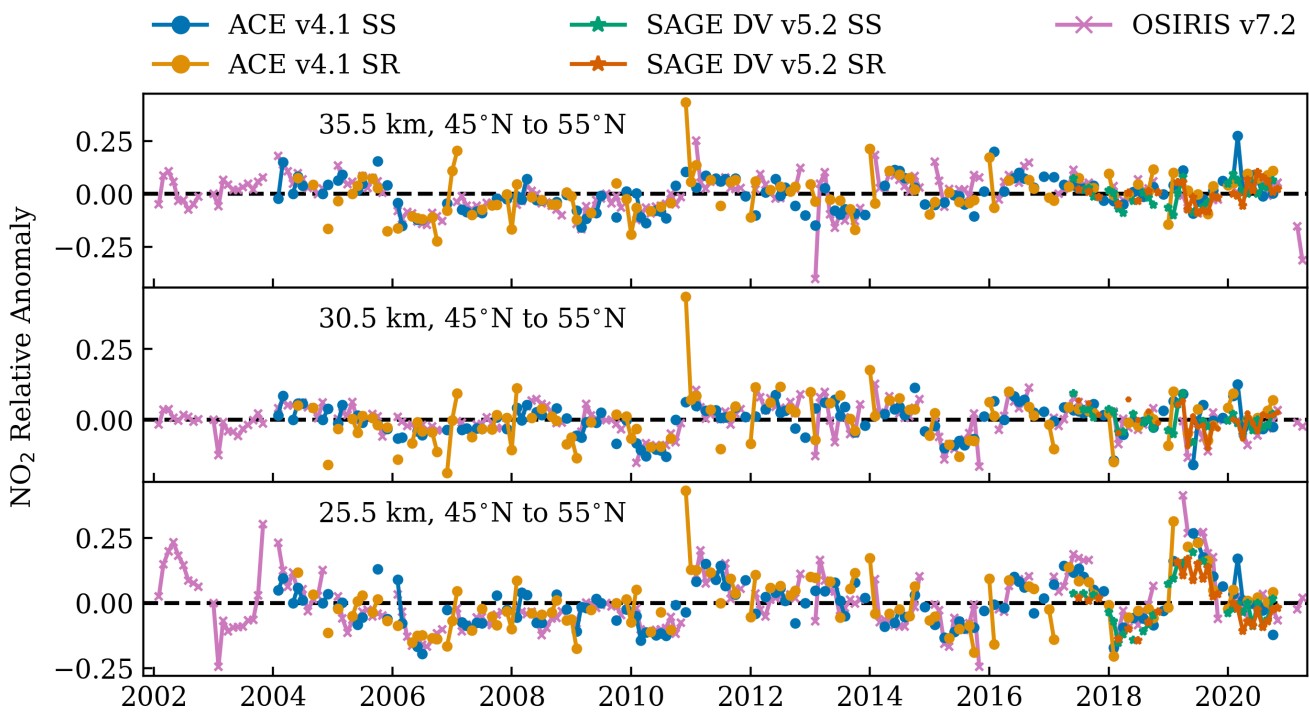

**Figure A3.** Anomaly time series comparing OSIRIS v7.2 to ACE v4.1 and SAGE III/ISS v5.2 DV for three altitudes at NH mid-latitudes.

Bognar, K., Tegtmeier, S., Bourassa, A., Roth, C., Warnock, T., Zawada, D., and Degenstein, D.: Stratospheric ozone trends for 1984–2021 in the SAGE II – OSIRIS – SAGE III/ISS composite dataset, Atmospheric Chemistry and Physics Discussions, 2022, 1–26, https://doi.org/10.5194/acp-2022-252, 2022.

Boone, C., Bernath, P., Cok, D., Jones, S., and Steffen, J.: Version 4 retrievals for the atmospheric chemistry experiment Fourier transform spectrometer (ACE-FTS) and imagers, Journal of Quantitative Spectroscopy and Radiative Transfer, 247, 106 939,
https://doi.org/https://doi.org/10.1016/j.jqsrt.2020.106939, 2020.

Boone, C. D., Nassar, R., Walker, K. A., Rochon, Y., McLeod, S. D., Rinsland, C. P., and Bernath, P. F.: Retrievals for the atmospheric chemistry experiment Fourier-transform spectrometer, Appl. Opt., 44, 7218–7231, https://doi.org/10.1364/AO.44.007218, 2005.

Boone, C. D., Walker, K. A., and Bernath, P. F.: Version 3 retrievals for the atmospheric chemistry experiment Fourier transform spectrometer (ACE-FTS), in: The Atmospheric Chemistry Experiment ACE at 10: a Solar Occultation Anthology, edited by Bernath, P., vol. 10, pp.
103–127, A. Deepak Publishing, 2013.

Bourassa, A. E., Degenstein, D. A., and Llewellyn, E. J.: SASKTRAN: A spherical geometry radiative transfer code for efficient estimation of limb scattered sunlight, Journal of Quantitative Spectroscopy and Radiative Transfer, 109, 52–73, https://doi.org/https://doi.org/10.1016/j.jqsrt.2007.07.007, 2008.

Bourassa, A. E., McLinden, C. A., Sioris, C. E., Brohede, S., Bathgate, A. F., Llewellyn, E. J., and Degenstein, D. A.: Fast $NO_2$ retrievals from Odin-OSIRIS limb scatter measurements, Atmospheric Measurement Techniques, 4, 965–972, https://doi.org/10.5194/amt-4-965-2011, 2011.

Brion, J., Chakir, A., Daumont, D., Malicet, J., and Parisse, C.: High-resolution laboratory absorption cross section of O3. Temperature effect, Chemical physics letters, 213, 610–612, https://doi.org/https://doi.org/10.1016/0009-2614(93)89169-I, 1993.

Brohede, S. M., Haley, C. S., Mclinden, C. A., Sioris, C. E., Murtagh, D. P., Petelina, S. V., Llewellyn, E. J., Bazureau, A., Goutail, F., Randall, C. E., et al.: Validation of Odin/OSIRIS stratospheric NO2 profiles, Journal of Geophysical Research: Atmospheres, 112, https://doi.org/https://doi.org/10.1029/2006JD007586, 2007.

Cisewski, M., Zawodny, J., Gasbarre, J., Eckman, R., Topiwala, N., Rodriguez-Alvarez, O., Cheek, D., and Hall, S.: The Stratospheric Aerosol and Gas Experiment (SAGE III) on the International Space Station (ISS) Mission, in: Sensors, Systems, and Next-Generation Satellites XVIII, vol. 9241, p. 924107, International Society for Optics and Photonics, https://doi.org/10.1117/12.2073131, 2014.

Daumont, D., Brion, J., Charbonnier, J., and Malicet, J.: Ozone UV spectroscopy I: Absorption cross-sections at room temperature, Journal of Atmospheric Chemistry, 15, 145–155, https://doi.org/10.1007/BF00053756, 1992.

Dubé, K., Randel, W., Bourassa, A., Zawada, D., McLinden, C., and Degenstein, D.: Trends and Variability in Stratospheric NOx Derived From Merged SAGE II and OSIRIS Satellite Observations, Journal of Geophysical Research: Atmospheres, 125, e2019JD031 798, https://doi.org/https://doi.org/10.1029/2019JD031798, 2020.

Dubé, K., Bourassa, A., Zawada, D., Degenstein, D., Damadeo, R., Flittner, D., and Randel, W.: Accounting for the photochemical variation in stratospheric $NO_2$ in the SAGE III/ISS solar occultation retrieval, Atmospheric Measurement Techniques, 14, 557–566, https://doi.org/10.5194/amt-14-557-2021, 2021.

Haley, C. S. and Brohede, S.: Status of the Odin/OSIRIS stratospheric O3 and NO2 data products, Canadian Journal of Physics, 85, 1177–1194, https://doi.org/10.1139/p07-114, 2007.

Haley, C. S., Brohede, S. M., Sioris, C. E., Griffioen, E., Murtagh, D. P., McDade, I. C., Eriksson, P., Llewellyn, E. J., Bazureau, A., and Goutail, F.: Retrieval of stratospheric O3 and NO2 profiles from Odin Optical Spectrograph and Infrared Imager System (OSIRIS) limb-scattered sunlight measurements, Journal of Geophysical Research: Atmospheres, 109, https://doi.org/10.1029/2004JD004588, 2004.

Llewellyn, E. J., Lloyd, N. D., Degenstein, D. A., Gattinger, R. L., Petelina, S. V., Bourassa, A. E., Wiensz, J. T., Ivanov, E. V., McDade, I. C., Solheim, B. H., McConnell, J. C., Haley, C. S., von Savigny, C., Sioris, C. E., McLinden, C. A., Griffioen, E., Kaminski, J., Evans, W. F., Puckrin, E., Strong, K., Wehrle, V., Hum, R. H., Kendall, D. J., Matsushita, J., Murtagh, D. P., Brohede, S., Stegman, J., Witt, G., Barnes, G., Payne, W. F., Piché, L., Smith, K., Warshaw, G., Deslauniers, D. L., Marchand, P., Richardson, E. H., King, R. A., Wevers, I., McCreath, W., Kyrölä, E., Oikarinen, L., Leppelmeier, G. W., Auvinen, H., Mégie, G., Hauchecorne, A., Lefèvre, F., de La Nöe, J., Ricaud, P., Frisk, U., Sjoberg, F., von Schéele, F., and Nordh, L.: The OSIRIS instrument on the Odin spacecraft, Canadian Journal of Physics, 82, 411–422, https://doi.org/10.1139/p04-005, 2004.

Malicet, J., Daumont, D., Charbonnier, J., Parisse, C., Chakir, A., and Brion, J.: Ozone UV spectroscopy. II. Absorption cross-sections and temperature dependence, Journal of atmospheric chemistry, 21, 263–273, https://doi.org/https://doi.org/10.1007/BF00696758, 1995.

McLinden, C. A., Olsen, S. C., Hannegan, B., Wild, O., Prather, M. J., and Sundet, J.: Stratospheric ozone in 3-D models: A simple chemistry and the cross-tropopause flux, Journal of Geophysical Research: Atmospheres, 105, 14 653–14 665, https://doi.org/10.1029/2000JD900124, 2000.

McLinden, C. A., Haley, C. S., and Sioris, C. E.: Diurnal effects in limb scatter observations, Journal of Geophysical Research: Atmospheres, 111, 2006.

Murtagh, D., Frisk, U., Merino, F., Ridal, M., Jonsson, A., Stegman, J., Witt, G., Eriksson, P., Jiménez, C., Megie, G., Noë, J. d. l., Ricaud, P., Baron, P., Pardo, J. R., Hauchcorne, A., Llewellyn, E. J., Degenstein, D. A., Gattinger, R. L., Lloyd, N. D., Evans, W. F., McDade, I. C., Haley, C. S., Sioris, C., Savigny, C. v., Solheim, B. H., McConnell, J. C., Strong, K., Richardson, E. H., Leppelmeier, G. W., Kyrölä, E., Auvinen, H., and Oikarinen, L.: An overview of the Odin atmospheric mission, Canadian Journal of Physics, 80, 309–319, https://doi.org/10.1139/p01-157, 2002.

Prather, M. and Jaffe, A. H.: Global impact of the Antarctic ozone hole: Chemical propagation, Journal of Geophysical Research: Atmospheres, 95, 3473–3492, https://doi.org/10.1029/JD095iD04p03473, 1990.

Rieger, L. A., Zawada, D. J., Bourassa, A. E., and Degenstein, D. A.: A Multiwavelength Retrieval Approach for Improved OSIRIS Aerosol Extinction Retrievals, Journal of Geophysical Research: Atmospheres, 124, 7286–7307, https://doi.org/10.1029/2018JD029897, 2019.

SAGE III Algorithm Theoretical Basis Document, T.: SAGE III Algorithm Theoretical Basis Document (ATBD) Solar and Lunar Algorithm, Tech. rep., LaRC 475-00-109, https://eospso.gsfc.nasa.gov/sites/default/files/atbd/atbd-sage-solar-lunar.pdf, 2002.

Sheese, P. E., Walker, K. A., Boone, C. D., McLinden, C. A., Bernath, P. F., Bourassa, A. E., Burrows, J. P., Degenstein, D. A., Funke, B., and Fussen, D.: Validation of ACE-FTS version 3.5 NOy species profiles using correlative satellite measurements, Atmospheric Measurement Techniques, 9, https://doi.org/https://doi.org/10.5194/amt-9-5781-2016, 2016.

Sioris, C. E., Haley, C. S., McLinden, C. A., von Savigny, C., McDade, I. C., McConnell, J. C., Evans, W. F. J., Lloyd, N. D., Llewellyn, E. J., Chance, K. V., Kurosu, T. P., Murtagh, D., Frisk, U., Pfeilsticker, K., Bösch, H., Weidner, F., Strong, K., Stegman, J., and Mégie, G.: Stratospheric profiles of nitrogen dioxide observed by Optical Spectrograph and Infrared Imager System on the Odin satellite, Journal of Geophysical Research: Atmospheres, 108, https://doi.org/https://doi.org/10.1029/2002JD002672, 2003.

Sioris, C. E., Rieger, L. A., Lloyd, N. D., Bourassa, A. E., Roth, C. Z., Degenstein, D. A., Camy-Peyret, C., Pfeilsticker, K. E., Berthet, G., Catoire, V., Goutail, F., Pommereau, J.-P., and Mclinden, C. E.: Improved OSIRIS NO2 retrieval algorithm: description and validation, Atmospheric Measurement Techniques, 10, 1155 – 1168, https://doi.org/10.5194/amt-10-1155-2017, 2017.

Vandaele, A. C., Hermans, C., Simon, P. C., Carleer, M., Colin, R., Fally, S., Merienne, M.-F., Jenouvrier, A., and Coquart, B.: Measurements of the NO2 absorption cross-section from 42 000 cm- 1 to 10 000 cm- 1 (238–1000 nm) at 220 K and 294 K, Journal of Quantitative Spectroscopy and Radiative Transfer, 59, 171–184, https://doi.org/https://doi.org/10.1016/S0022-4073(97)00168-4, 1998.

Zawada, D., Dueck, S., Rieger, L., Bourassa, A., Lloyd, N., and Degenstein, D.: High-resolution and Monte Carlo additions to the SASK-TRAN radiative transfer model, Atmospheric Measurement Techniques, 8, 2609–2623, https://doi.org/https://doi.org/10.5194/amt-8-2609-2015, 2015.