# Peer review of "An Improved OSIRIS NO2 Profile Retrieval in the UTLS and Intercomparison with ACE-FTS and SAGE III/ISS"

_Atmospheric Measurement Techniques, 2022_

## Author Comment (AC2)

[Figure]

Figure 1: Decrease in the number of coincident profiles with OSIRIS after removing OSIRIS profiles with a SZA > 85.

[Figure]

Figure 2: Difference between the bias with and without including OSIRIS scans with SZA < 85. Top left: ACE-FTS sunrise occultations. Top right: SAGE III/ISS sunrise occultations. Bottom left: ACE-FTS sunset occultations. Bottom right: SAGE III/ISS sunset occultations. In all panels the dashed line is the average tropopause altitude, and the dotted line is the average 380 K potential temperature altitude.

---

## Author Response (AR1)

**Response to Reviewers**

Thank you for taking the time to review our manuscript and for providing helpful suggestions. Our responses to each comment are included here.

Anonymous Referee #1

This paper presents the new version (v7.2) of the OSIRIS $NO_2$ retrieval. Compared to version v6.0, the sensitivity in the UTLS is improved and the previously observed low bias is reduced. Good agreement (within 20%) is found with ACE/FTS and SAGE III/ISS data sets. The OSIRIS $NO_2$ monthly zonal mean data also show a variability in time which is very similar to ACE/FTS and SAGE III/ISS at most altitudes and latitudes.

This study fits well with the scope of AMT and the manuscript is well written and clearly structured. I recommend publishing the paper in AMT after addressing the following specific comments:

Page 2, Section 2.2: This Section is a bit difficult to follow. I would recommend the authors to summarize all those validation results in a table including the following entries (or something similar): OSIRIS retrieval version, ancillary data + version, local time/SZA used for the comparison, altitude range, comparison results.

 A table summarizing these results has been added to the manuscript.

Page 3, Section 2.3: The authors should describe in the first paragraph how the temperature dependences of the O3 and $NO_2$ cross-sections are treated in their retrieval. Also, are the absorptions by O4 and water vapour included in the retrieval? Again, a table summarizing the main retrieval settings could be helpful for the reader here.

 The temperature dependence is handled in two different ways.  The forward model radiative transfer calculation includes the full temperature dependence at all altitudes.  For the regression for each line of sight we use the temperature at the tangent point to compute the cross section. This is now mentioned in the manuscript. A table of the main retrieval settings has also been added to the manuscript

No, the absorptions by O4 and water vapour are not included in the retrieval. Both water vapour and O2-O2 are spectrally uncorrelated with $NO_2$ and weakly absorbing in the spectral region used, so we do not expect biases from neglecting them.  However, it is something to consider for future data versions.

Page 4, lines 80-82: The threshold value of 1.01 should be justified.

 We have added the statement "which results in profiles that have converged to a level that is orders of magnitude less than the estimated precision".  The 1.01 here is actually a very strict convergence criteria for the values it is testing, it is not comparable to something like a 1% chance in chi2 between iterations which would be a much weaker criterion.

Page 5, lines 112-113 and page 6, lines 126-127: the new retrieval version allows to retrieve negative number density values. Is any quality-control criterion applied on these profiles with negative values? For instance, do you reject profiles with negative values which are below a given threshold?

Negative values are not filtered out, and in fact should be used in computing quantities such as a monthly zonal mean or in the comparisons presented here or else the results will be biased high. This is now mentioned in the manuscript.

Figure 3, page 7: Why the 10°S-30°S and 10°N-30°N latitude bands are not considered in this figure? Also, nothing is said about the probability densities of both retrieval versions at high (>50°) latitudes. If not included in Figure 3, both aspects should be at least discussed here. Another option would be to put the figures with the missing latitude bands in an annex.

The figure has been updated to include all latitudes.

Figure 4(a): Why not showing an example of averaging kernel peaking at an altitude below 15km, i.e. with a difference between the nominal altitude and the altitude derived from the Gaussian fit which is larger than the threshold value of 1.5km? This would better illustrate your kernel filter approach.

Thank you for the suggestion, the example altitude in the figure has been changed to 15.5 km.

Page 11, lines 216-227: the photochemical correction applied to all data sets and which consists in shifting all of them to 12:00 pm is a critical point and to my opinion, the uncertainty associated to this correction should be better characterized. The 1% uncertainty on NO2 obtained by perturbing the main input of the model is likely correct but this is clearly a lower estimate of the photochemical correction uncertainty. In order to get a better estimate, I recommend to make some sensitivity tests on the rate constants (and their respective uncertainties) of the main reactions involving NO2. Also, nothing is said about the stratospheric aerosols. Did you include them in your photochemical box model simulations? If yes, are they those simultaneously retrieved from the OSIRIS measurements (see page 3, lines 67-70) ?

Thank you for the suggestion. It would be worthwhile to test the effect of varying the rate constants in PRATMO, however it is not feasible to do within the scope of this study as the PRATMO software is not designed to have the rate constants altered. It is also worth noting that the reaction rates used PRATMO are the same as those used by larger models like WACCM, so they are likely well quantified (the rate constants are taken from Burkholder et al (2015)).

We do not include aerosols in the box model calculations. It would be interesting to investigate their effect in the future.

Burkholder, J. B., Sander, S. P., Abbatt, J. P. D., Barker, J. R., Huie, R. E., Kolb, C. E., ... & Wine, P. H. (2015). *Chemical kinetics and photochemical data for use in atmospheric studies: evaluation number 18*. Pasadena, CA: Jet Propulsion Laboratory, National Aeronautics and Space Administration, 2015.

Figure 7, page 12: Why not including also mid-latitude bands?

Mid latitudes have been added to the figure.

Page 13, line 251-252: The lower bias in the SH is attributed to the sampling of coincident profiles. I think this point should be further discussed in a quantitative way, i.e. how different are the SH and NH samplings?

This is just a theory because each latitude is sampled mostly in specific months. We have removed the statement from the manuscript. The hemispheric difference seems to just be a feature of the ACE data as it does not appear in the coincidences with SAGE. The hemispheric difference is also present in the

comparisons between both OSIRIS v6.0 and v7.2 with ACE. As the difference is not caused by the OSIRIS retrieval, determining its origin is outside the scope of this work.

Page 14, lines 255-257 + Figure 9: The application of the kernel filter can have a huge impact on the retrieval results below the tropopause. What is at the end the official v7.2 product? Is it with or without applying this filter?

The official v7.2 product includes a variable for the averaging kernel filter altitude so that users can apply it if desired. This is now mentioned in the conclusion of the manuscript.

Figure 12, page 17: How did you select the altitude and latitude ranges shown in this figure? Are they representative of other altitude and latitude ranges? Maybe you could show the plots for all the altitude/latitude range combinations in an annex? This could be useful for those readers interested in stratospheric NO2 trend analysis.

We chose bins that are generally representative, but also focused on the lower altitudes as this is where the OSIRIS retrieval changed the most. Several more bins have been added to an appendix for the interested reader.

Technical corrections:

The date format is not consistent throughout the manuscript (e.g. we can find 06:30 am, 06:30 a.m, 06:30 AM). Please check.

This has been fixed.

Chris Mclinden

This is a solid paper on a topic that fits in AMT. It describes a new OSIRIS NO2 data product (version 7.2) that improves upon previous versions, addressing deficiencies in the retrieval methodology such as how the reference radiance is calculated and changes in spectral resolution with instrument temperature. Comparisons are made with co-incident observations from occultation instruments. It is well written and the subject is worthy of publication in AMT.

My only criticism of substance is that considering "… version 7.2, was designed to fix a low bias and to improve performance in the UTLS …" it really does not do an adequate job evaluating the performance of the new product in this region. This paper should attempt to better answer questions such as "precisely how good is v7.2 in the UTLS" or "is v7.2 actually better than the old version", and if this cannot be answered, then why not. This is the region where diurnal effect errors can be quite substantial. Of the three datasets used, it was ignored in OSIRIS and ACE-FTS, and corrected for in ISS/SAGEIII, which muddies the waters. For example, the OSIRIS-ACE and OSIRIS-SAGE results seem to conflict. Does that mean there are biases between ACE and SAGEIII? Is it related to the diurnal effect? I made some suggestions below as to how to investigate the diurnal effect as, especially for OSIRIS, it is not that easy. That said, the lead author is well equipped to examine this further. It is also difficult to know what to make of the comparison that go all the way down an altitude of 10 km. Is there sufficient data for these to be meaningful? And does the variability swamp the signal? All-in-all this part of the paper needs additional analysis such that they would support some clear summary findings.

Once this, along with several smaller clarifications outlined below, is completed I would support publication.

Thank you for the thoughtful response. We have performed further analysis to determine the effect of diurnal variations on the intercomparisons presented in our paper. Several paragraphs discussing these results, along with a discussion of why we think v7.2 is an improvement over v6, have been added to the conclusion of our manuscript:

NO2 from the SAGE III/ISS DV retrieval shows improved agreement with OSIRIS compared to NO2 from standard SAGE retrieval. The diurnal effect produces a high bias in NO2 retrieved from occultation instruments below 25 km. There is no version of the ACE-FTS NO2 retrieval that accounts for diurnal variations along the line of sight, which could be increasing the difference with OSIRIS below about 25 km. It is not expected that the diurnal effect would be greater in ACE-FTS than in SAGE III/ISS, so this will add at most a 5% to 40% bias, with the largest bias at the lowest altitudes (Dube et al 2021).  If we assume an approximate high bias of 25% below 20 km in the ACE-FTS NO2, the difference between ACE-FTS and OSIRIS will become greater near the tropopause, and especially in the NH. The bias in this region will be >50%, as it is for SAGE III/ISS and OSIRIS. In the SH the bias between ACE-FTS and OSIRIS is ~ -15% below 20 km. In this region including the diurnal correction will result in a bias of ~10%, corresponding to improved agreement between OSIRIS and ACE-FTS. This is opposite in sign and smaller in absolute value than the bias between SAGE III/ISS and OSIRIS in this region.  There are significantly more coincidences between ACE-FTS and OSIRIS than there are between SAGE III/ISS and OSIRIS at SH mid-high latitudes, which could explain why we see a smaller bias between ACE-FTS and OSIRIS.

We also considered the effect of diurnal variations along the OSIRIS LOS by repeating the analysis using only OSIRIS scans with a SZA <85, corresponding to scans shown by McLinden et al (2006) to have a significant bias. Removing these scans had a minimal effect on the comparisons between OSIRIS and ACE-FTS in the stratosphere, changing the bias by at most 3%. It had a greater effect on the comparison with SAGE III/ISS, where the agreement improved by up to 10% in some bins in the lower stratosphere.

Overall, we conclude based on comparison with NO from ACE-FTS and SAGE III/ISS that the OSIRIS v7.2 NO2 product is an improvement over the previous v6.0 OSIRIS NO2, particularly below 30 km. The bias between OSIRIS v7.2 and both ACE-FTS and SAGE III/ISS has decreased from v6.0, except for in a few bins near the tropopause where including the diurnal effect in SAGE III/ISS (or estimating the magnitude of the effect for ACE-FTS) makes the bias greater than it was in v6.0

Line 20: you spelled out the acronyms in the abstract, but do so when first used in the main test as well.

The AMT style guide requires defining acronyms in both the abstract and the main text (https://www.atmospheric-measurement-techniques.net/submission.html#english)

Line 27-28:  use "… PRATMO stratospheric photochemical …" and " … McLinden et al. (2000) and later Adams et al. (2017)." (https://doi.org/10.5194/acp-17-8063-2017)

This reference has been added

Figure 2:  what time period is this for?

It is from November 2001 to May 2020. This is now mentioned in the figure caption.

Line 95: "worse agreement between the OSIRIS measurements and the forward model" Worse how? Shouldn't this be one of the parameters one can optimized by simulating OSIRIS radiances accounting for precision and sampling/field of view considerations. The 2-3 km over most of the stratosphere argument would seem to apply for any of the values of alpha used. Would a smaller alpha help in the UTLS?

 We have added ", particularly above 30 km." to be clear what this specific statement was referring to.

Generally ideal retrieval theory would say that reducing alpha increases sensitivity, at the cost of poorer precision. This is something that could be simulated as the reviewer suggests. However we have noticed that the retrieval tends to behave poorly at low altitudes for smaller values of alpha, and in particular fails to converge often. This is likely due to an effect where small biases in the forward model (or assumptions made) are amplified as the amount of regularization decreases. The biases are things that are challenging to simulate such as patchy clouds, or even identify in some cases. Essentially what we try to do is pick the lowest value of alpha where the retrieval converges most of the time and still behaves reasonably.

Line 110:  Be more specific about where v6 and v7.2 begin to differ.  E.g. do they both solve for the minimum in equation

 A table has been added which summarizes the main features of each retrieval version.

Line 125: "The log-normal distributions are less physically realistic" … why is this?  This could be true but maybe it is not that obvious.  Please give some rationale for this.  If this is in the upper troposphere one might expect something non-Gaussian for if there is occasional lightning-NOx.

 This section was a little unclear.  At the lowest altitudes these distributions are actually dominated by the precision of the measurements rather than geo-physical NO2 variations which is why we expect them to be normally distributed.  We have added a statement to that effect.

Line 139 or thereabouts:  what is the DOFS, or range in DOFS, for this data product?

The value depends on the profile, but it is typically around 13. However, we prefer to use vertical resolution as the diagnostic because the DOFS depends on the lowerbound altitude.

Line 144: please motivate the use of 1.5 km as the threshold a little more.

The larger the value the less data is removed. We tested several values for the threshold and 1.5 km was a good compromise between removing what is likely good data and not removing what is likely bad data. A larger difference means we are including information that is quite far from the tangent point. This is now mentioned in the manuscript.

Line 155: "… kernel filter, as used in version 6.2"?

 The kernel filter is applied to the version 7.2 data. This is now mentioned in the manuscript.

Line 170: "change in the processor"… what does this mean? Same for "global environment settings" later on.

This means that the processing was moved to a different computer.

Line 181:  would the lower altitude of the ISS, relative to SCISAT, mean there is ~1 more orbit per day... I think there are generally 16 orbits per day for the ISS

You are correct, this has been fixed.

Line 192: does OSIRIS not need to worry about O4 absorption in their NO2 retrieval?

The inclusion of O2-O2 absorption typically makes very little difference in the NO2 spectral region since it is largely uncorrelated with the NO2 absorption.  For SAGE-III it likely makes a larger difference for the O3 retrieval.

Line 203: So this is an off-line (unofficial) retrieval, correct?  This should be clarified.  Related: later on you use "SAGE III/ISS DV v5.2"... what does this mean?  It implies, I think, that the product is v5.2 but with a diurnal correction.  If you did your own retrieval then I would not use the 'v5.2' label which implies to me that it is an official product.

 The correction is applied to the official v5.2 data product. This is what we mean to suggest by the label DV v5.2. We have clarified this in the manuscript.

Line 221-224: this is good information to have in section 3.2.1

 This information has been moved to section 3.2.1

Line 225: "The effect of changes in the input parameters on the PRATMO NO2 was estimated by perturbing them in the model."  What was this used for?

This information was not used directly, it is just included to provide an idea of how sensitive the modelled NO2 is to the input parameters.

Line 228-229: "... it is not always enough to account for the difference between sunrise and sunset occultations from a single instrument."  Is the purpose of this statement to point out that systematic errors may still remain suggesting that the scaling is not perfect?  My guess is that there could be sampling issues that would also cause the SR and SS to not be the same anyway.

 Yes, we are just suggesting that the scaling is not perfect. You are correct, the sampling differences likely also contribute.

Figure 7: "... coincident NO2, when shifted from their individual local times to 12:00 pm, for several altitudes..."   is this what you mean?  If not, please clarify.

 Yes, this is what we mean. The wording has been adjusted.

Line 245: Shouldn't v7 have more data lower down based on Figure 5?

Figure 5 is comparing the amount of v7.2 data retrieved before and after the application of the averaging kernel filter. The combination of this filter, plus the improved cloud filtering in v7.2, means that we are removing more (unrealistic) data from v7.2 than from v6 at the low altitudes.

Line 245: showing the standard deviations of the average differences, or the average ± s.d., would suggest if this is due to fewer coincidences or not.

It does not really make sense to compare the standard deviations at these lower altitudes because the data is normally distributed in v7.2, but has closer to a lognormal distribution in v6.0.

Section 4.1: any discussion of the differences between ACE and OSIRIS below 20 km needs to factor in diurnal effect errors. Neither ACE nor OSIRIS corrects for them, but you should be able to deduce the impact of these based on your work on ISS/SAGE III and other publications on the subject. Previous comparisons, e.g., see Brohede et al. (doi:10.1029/2006JD007586) Figure 9, show the expected high bias in ACE-FTS. The effect in OSIRIS is harder to generalize due to the varying geometry, but was estimated in McLinden et al. (doi:10.1029/2005JD006628) Figures 4 and 5. You might consider limiting OSIRIS to SZA<85 OR [SZA<85 or (SZA<88 and 75<dAZ<105)]

Thank you for the suggestion. We have further investigated the diurnal effect in ACE and OSIRIS NO2. Requiring the OSIRIS scans to have SZA < 85 reduced the number of coincidences with ACE by ~14%. Similarly, there are ~12% fewer coincidences between SAGE III/ISS and OSIRIS. Most of the lost coincidences are in the SH. It is worth noting that we only retrieve data for scans with SZA < 88 to begin with.

[Figure]

We redid the comparisons after removing these coincidences and found that the effect is generally negligible. Removing these coincidences changes the bias between OSIRIS and ACE in the stratosphere by at most ~+/-3%. For SAGE III/ISS the effect is larger in a few bins, particularly near 50 S for the SS occultations (which is the bin that loses the most profiles after filtering).

The following figure compares the difference between the mean bias in each bin with and without including scans where OSIRIS has an SZA>85. The top row is for sunrise occultations, and the bottom row is at sunset. A negative value shows that the bias with OSIRIS was reduced after removing scans with a significant diurnal effect.

[Figure]

Based on this analysis we conclude that the diurnal effect in OSIRIS data has a negligible effect on the comparison with ACE-FTS in the stratosphere, and a significant effect on the comparison with SAGE, but only in a few bins.

The diurnal effect will have a greater impact on ACE NO2 than it does on OSIRIS NO2. From Dube et al 2021 and Brohede et al 2007 we expect there to be a high bias of greater than 10% in the ACE NO2 below ~30 km. Accounting for this effect would improve the agreement with OSIRIS in the SH, but decrease agreement in the NH where ACE is already low compared to OSIRIS.

A section discussing these findings has been added to the manuscript (section 4.3).

Section 4.2:  What is the impact of the diurnal correction on the SAGEIII profiles?  It is a little hard to tease in the figure.  What do you make of the OS-S3 comparisons vs the OS-ACE below 20 km?

The impact of the diurnal correction on SAGE III is described in detail in Dube et al 2021(https://amt.copernicus.org/articles/14/557/2021/). Including this correction reduces the SAGE III NO2 by about 10% at 25 km and more than 10% at lower altitudes (see figure 7 of Dube et al 2021). This is now mentioned in the manuscript.

Below 20 km we expect the ACE NO2 to be biased high in the same way that the SAGE III NO2 was before being corrected for diurnal effects. If we assume an approximate high bias of 25% in the ACE NO2, the difference between ACE and OSIRIS will become greater near the tropopause, and especially in the NH. The bias in this region will be >50%, as it is for SAGE III and OSIRIS. In the SH the bias between ACE and OSIRIS is ~-15% below 20 km. In this region including the diurnal correction will result in a bias of ~10%, corresponding to improved agreement between OSIRIS and ACE. This is opposite in sign and smaller in absolute value than the bias between SAGE and OSIRIS in this region. There are significantly

more coincidences between ACE and OSIRIS than there are between SAGE and OSIRIS at SH mid-high latitudes, which could explain why we see a smaller bias between ACE and OSIRIS. We have added this discussion to the conclusion of the manuscript.

Line 274: "…from a data set…" what does this mean?

We intended to say "The relative anomaly is calculated by subtracting the overall mean value for a given month from all values for that month to remove the seasonal cycle (e.g. the mean June NO2 concentration is subtracted from each individual June NO2 concentration)". This has been corrected in the manuscript.

Line 284-285: "At sunrise ACE-FTS samples a region of the atmosphere that has not been illuminated long enough for the NO2 to reach equilibrium, however this is not an issue at sunset." Are you agreeing this is true? I am not sure it is. Or are you merely restated the argument from Sheese et al.? Neither would be in a pseudo-steady-state at the terminator.

We are restating the theory of Sheese et al (2016)

Line 291: was the trend removed in these timeseries?

No, the trend was not removed. The only thing we removed was the seasonal cycle.

Line 316: "ACE-FTS retrieval it would likely improve agreement with OSIRIS" … would it not make ACE-FTS values larger, thereby increasing the bias? I agree the diurnal effect will be similar for ACE and SAGEIII, and thus, either way you could be more quantitative here.

The ACE values would become smaller if the diurnal effect was included. We have elaborated on the discussion of the diurnal effect in the conclusion of the manuscript.